# Mechanisms of soil organic carbon and nitrogen stabilization in mineral associated organic matter – Insights from modelling in phase space

Stefano Manzoni[1], Francesca Cotrufo[2]

[1]Department of Physical Geography and Bolin Centre for Climate Research, Stockholm University, SE-106 91 Stockholm, Sweden

[2]Department of Soil and Crop Sciences, Colorado State University, Fort Collins, CO, USA

*Correspondence to*: Stefano Manzoni (stefano.manzoni@natgeo.su.se)

**Abstract.** Understanding the mechanisms of plant-derived carbon (C) and nitrogen (N) transformation and stabilization in soil is fundamental for predicting soil capacity to mitigate climate change and support other soil functions. The decomposition of plant residues and particulate organic matter (POM) contributes to the formation of mineral associated (on average more stable) organic matter (MAOM) in soil. MAOM is formed from the binding of dissolved organic matter (ex vivo pathway) or microbial necromass and bioproducts (in vivo pathway) to minerals and metal colloids. Which of these two soil organic matter (SOM) stabilization pathways is more important and under which conditions remains an open question. To address this question, we propose a novel diagnostic model to describe C and N dynamics in MAOM as a function of the dynamics of residues and POM decomposition. Focusing on relations among soil compartments (i.e., modelling in phase space) rather than time trajectories allows isolating the fundamental processes underlying stabilization. Using this diagnostic model in combination with a database of 36 studies in which residue C and N were tracked into POM and MAOM, we found that MAOM is predominantly fuelled by necromass produced by microbes decomposing residues and POM—the so-called 'in vivo' pathway of stabilization. The relevance of the in vivo pathway is higher in clayey soils, but lower in C rich soils and with N poor added residues. Overall, our novel modelling in phase space proved to be a sound diagnostic tool for the mechanistic investigation of soil C dynamics and supported the current understanding of the critical role of both microbial transformation and mineral capacity for the stabilization of C in mineral soils.

## 1. Introduction

Soil carbon (C) storage has been proposed as a climate mitigation strategy, but how much C can be stored in soil and for how long is a matter of debate. Increasing plant productivity or adding C amendments to soils can increase C stocks or slow down their decline (Bruni et al., 2022), but the persistence of the added C depends on the balance of stabilization and destabilization processes (Lehmann et al., 2020; Liang et al., 2017). Only a small fraction of the added C is retained in the soil in the long term in mineral associated forms or occluded in stable aggregates (Cotrufo et al., 2015; Manzoni et al., 2018; Pries et al., 2017). Yet, even small annual increments in soil C stocks over large areas can support the climate mitigation effort—not to mention other benefits of organic matter-rich soils (Paustian et al., 2016). The clear advantages of promoting C storage in soil motivate an improved understanding of C stabilization pathways.

Here we focus on stabilization by mineral association including within fine aggregates (<53 µm) and do not consider occlusion in larger aggregates, partly because the stability of mineral associated organic matter (MAOM) is on average higher, and partly because of data availability. Two main pathways support mineral association of organic matter (Liang et al., 2017): i) the 'in vivo' pathway, in which microbial growth generates necromass and extracellular products that are stabilized on soil minerals, and ii) the 'ex vivo' pathway, in which low molecular weight compounds, released by the depolymerization of structural residues and particulate organic matter (POM) by extracellular enzymatic reactions or from root exudates, are stabilized on soil minerals. Both pathways are partly mediated by microbial (and faunal) decomposers. On the one hand, higher microbial growth per unit C consumed (i.e., high C use efficiency, CUE) is associated with higher necromass and thus to higher C storage—consistent with the in vivo pathway (Tao et al., 2023; Wang et al., 2021). On the other hand, higher microbial growth can lead to higher enzyme production, thereby promoting residue and soil organic matter (SOM) decomposition while also promoting C stabilization via the ex vivo pathway, with an uncertain net outcome on C storage.

While microbial growth, CUE and decomposition dynamics mediate C stabilization, ultimately in aerated mineral soils C is stabilized by association to soil minerals and amorphous metals. Therefore, their availability and capacity to interact with organic compounds set the potential for long-term C stabilization (Georgiou et al., 2022; Kögel-Knabner et al., 2008). Short-range ordered iron and aluminum oxides, and exchangeable calcium and magnesium promote organic matter stabilization by adsorption, as demonstrated by their strong correlations with MAOM (King et al., 2023). From a less mechanistic point of view, the clay (or silt + clay) fraction is also associated to higher proportion of MAOM in soil organic matter (Cotrufo et al., 2019), higher MAOM content (Begill et al., 2023), and faster stabilization of residue-derived C into MAOM (Haddix et al., 2020).

Also, the quality of the organic matter supplied to the soil plays a role in the C stabilization process. Residues rich in nutrients—especially nitrogen (N)—support microbial growth by providing microbes with a stoichiometrically balanced diet, thus resulting in higher CUE and ultimately higher likelihood of C stabilization in MAOM (Cotrufo et al., 2013). In contrast, microbes feeding on N-poor residues need to invest more resources in extracellular enzymes to mine nutrients and to release C in excess of their stoichiometric requirements, leading to lower CUE (Manzoni et al., 2017) and thus a less effective in vivo

pathway. Consistent with this idea that N-rich residues promote C stabilization in MAOM, residue N content and soil C stocks are positively correlated at regional scale (Zhou et al., 2019). However, mineral fertilizers can reduce the overall soil organic

matter stability by promoting C accumulation in particulate fractions with faster turnover (Rocci et al., 2022).

The combined effects of biota, soil properties, and input quality make prediction of C stabilization difficult, but these complexities are further compounded by methodological differences in the way organic matter fractions and their stability are identified. In general, organic matter is partitioned among still undecomposed coarse residues, particulate organic matter (POM) encompassing partly decomposed or fragmented residues (free or occluded in aggregates), and MOAM encompassing

more degraded compounds and necromass that are bound to soil minerals. These fractions are operationally defined in multiple ways—e.g., based on density or size fractionation; or considering sub-fractions occluded in aggregates or free (Leuthold et al., 2023). Moreover, due to nonlinear interactions of residues and native organic matter (priming), determining the fate of organic matter added to the soil as POM, MAOM, or mineralized products is possible only by tracing residue-derived C and N into the different soil components—e.g., through C and/or N isotope labelling. While soil fractionation combined with isotopic

labelling is a commonly employed methodology, it is laborious and, as a result, residue incorporation studies have low temporal resolution. Finally, the lack of common protocols makes comparisons across studies difficult. To overcome these methodological challenges, it can be useful to develop minimalist C and N dynamics models to be used as diagnostic tools to track residue C and N stabilization into MAOM.

A diagnostic model able to interpret observed C and N dynamics in residue, POM, and MAOM during decomposition can be

useful also to reconcile different trends that have been reported. In fact, residue-derived POM can both increase (Fulton-Smith and Cotrufo, 2019; Leichty et al., 2021) or decrease through time (Cheng et al., 2023; Neupane et al., 2023). Also MAOM can exhibit contrasting trends (increasing in the studies cited above, but decreasing in e.g., Wang et al., 2017). These contrasting temporal dynamics might be either the result of complex stabilization dynamics or—on the contrary—a consequence of different experimental approaches (e.g., residue placement above- or belowground) and sampling times across experiments

that mask simple underlying patterns.

We expect that the general pattern of stabilization is simple and universal—MAOM C and N increase as residues and POM are decomposed thanks to both in vivo and ex vivo pathways, but ultimately even MAOM C and N are mineralized—although it might take years to centuries and in some soils even millennia. We argue that this pattern would emerge clearly when modelling POM and MAOM dynamics not as a function of time, but in relation to each other—e.g., modelling variations in

MAOM as a function of variations in residues and POM. This approach moves away from classical modelling of time trajectories and focuses instead on modelling in the space of the state variables, also referred to as 'phase space' (Argyris et al., 1994). Phase space representations allow reducing the effects of factors that determine biogeochemical reaction rates (e.g., temperature, incubation conditions) while emphasizing instead the relations among the soil C and N compartments. For example, MAOM may accumulate in one dataset, but decrease in another. In the phase space, these two contrasting patterns

would appear as two subsequent phases in the same trajectory—as residues and POM are decomposed, MAOM first accumulates and then is depleted, forming a single humped-shaped trajectory in the MAOM vs. POM space. Similar phase

space representations have been applied to study nutrient dynamics during decomposition (Bosatta and Ågren, 1985; Manzoni et al., 2008), but to our knowledge they have not been used to investigate C and N stabilization mechanisms.

The goal of this contribution is to characterize the pathways of residue C and N stabilization using a novel, fully analytical diagnostic model combined with a database of 36 isotope labelling studies. Our specific questions are:

    i)        Can we reconcile contrasting patterns of residues and POM, and MAOM loss/accumulation by considering the dynamic coupling between these pools as decomposition progresses?

    ii)       What is the dominant pathway of C and N stabilization in MAOM?

    iii)      What are the drivers of the stabilization pathway as represented by model parameters?

## 2.   Methods

### 2.1. Theory

#### 2.1.1.   Model rationale

For the purpose of this model, we conceptualize soil organic matter as the sum of two physically well-defined compartments: combined residues and particulate organic matter (residues + POM, subscript *P*) and mineral associated organic matter

 (MAOM, subscript *M*). Residues + POM include both partly decomposed residues (operationally defined as fragments larger than 2 mm or separated by hand based on visual inspection) and organic matter in the light or coarse soil fractions (respectively isolated via density or size fractionation). The choice of merging residues and POM in one model compartment is motivated by the fact that in many datasets they were not separated. MAOM includes only organic matter in the heavy or fine soil fractions (also from density or size fractionation). Both compartments are characterized by their C and N contents (mass of C or N per

 unit soil dry mass). Moreover, different from other existing models, we consider microorganisms driving the decomposition process to be distinct for POM and MAOM, given the distinctive chemical and stoichiometric properties of these two soil compartments, though they might express similar traits so as to be functionally equivalent.

The two compartments are linked by two types of mass flow from residues + POM to MAOM: i) products of depolymerization of residues + POM transferred to MAOM in dissolved form before being converted into microbial biomass (ex vivo pathway

 of stabilization) and ii) necromass of microbes grown on residues + POM transferred to MAOM (in vivo pathway of stabilization). For simplicity, we do not consider dissolved organic matter (DOM) explicitly in this model. As shown in the Supplementary Information (Section S1), a model including DOM shared by microbes in both residues + POM and MAOM can be constructed, but this more general model can be approximated by the simpler one used here by making two assumptions: i) microbial uptake of the shared DOM is negligible compared to uptake from the depolymerization of residues + POM

 substrates, and ii) the DOM compartment is at quasi-equilibrium, which is a reasonable assumption because DOM is a relatively small pool with fast turnover time.

Model parameters allow regulating how C and N are partitioned between the two stabilization pathways. Moreover, we consider the possibility that the soluble fraction of the added residues is immediately stabilized as MAOM. Leaching of dissolved organic matter is neglected. In the datasets we used (Section 2.2.1), both C and N were added in the soil only at the

 beginning of the incubations, allowing us to track a single organic matter cohort. This means that the initial condition in the model represents how much C and N have been added, but there are no subsequent inputs. In natural conditions, there would also be continuous inputs from new residues incorporated in the soil and from root exudation—these inputs could be added to apply this model in other contexts. Carbon is lost through microbial respiration, while we do not track the fate of inorganic N accumulating due to net N mineralization.

 With this conceptual view of the soil system, we can write the mass balance equations for C (Section 2.1.2) and N (Section 2.1.4) of both substrate and microbial decomposers in the two compartments. These equations are not solved through time as usually done with this type of models, but instead we find analytically how one state variable change as a function of another

state variable (Sections 2.1.3 and 2.1.5). In other words, we solve the equations in 'phase space' (Argyris et al., 1994). Symbols are defined in Table 1, a schematic of the model is shown in Fig. 1, and a summary of equations for the model solution (including various scenarios with specific parameters) is provided in Table 2.

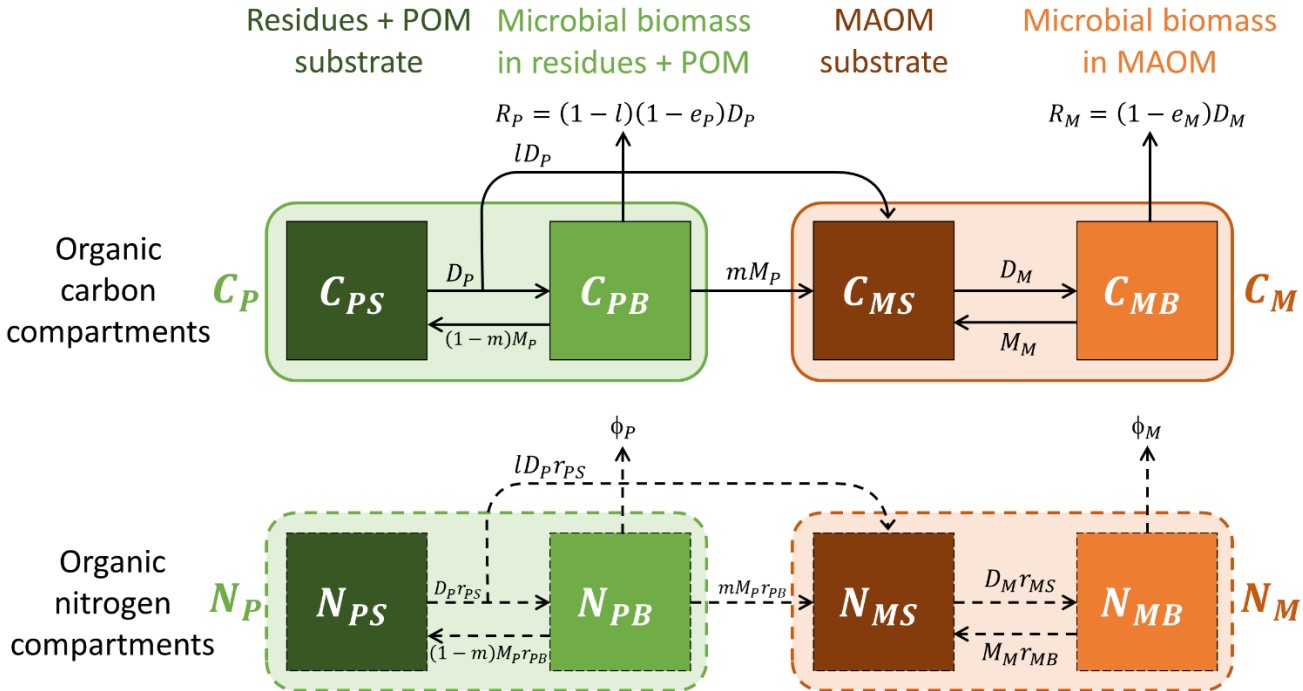

**Figure 1. Model schematic (see symbol explanations in Table 1). Solid and dashed arrows or compartment edges indicate respectively C and N flows or compartments. Light shading and color-coded symbols indicate aggregated variables including both substrates and microbial biomass. No input rates are shown because a single cohort of residues is tracked during decomposition and stabilization.**

**Table 1. Symbol definitions and units (see also Fig. 1). Subscripts $i=P$ and $M$ indicate state variables, rates, or parameters associated with plant residues and particulate organic matter (POM), and mineral associated organic matter (MAOM), respectively.**

| Symbol | Explanation | Units |
|---|---|---|
| State variables and independent variables | | |
| $c_i$ | Fraction of added residue C recovered in compartment $i$, $c_i = C_i/C_{P,0}$ | - |
| $C_i$ | Total C content in compartment $i$ | g C kg$^{-1}$ |
| $C_{iB}$ | C content in microbial biomass associated with compartment $i$ | g C kg$^{-1}$ |
| $C_{iS}$ | Substrate C content in compartment $i$ | g C kg$^{-1}$ |
| $n_i$ | Fraction of added residue N recovered in compartment $i$, $n_i = N_i/N_{P,0}$ | - |
| $N_i$ | Total N content in compartment $i$ | g N kg$^{-1}$ |
| $N_{iB}$ | N content in microbial biomass associated with compartment $i$ | g N kg$^{-1}$ |
| $N_{iS}$ | Substrate N content in compartment $i$ | g N kg$^{-1}$ |
| $r_i$ | N:C ratio of compartment $i$, $r_i = N_i/C_i$ | g N g C$^{-1}$ |
| $r_{iS}$ | N:C ratio of substrates in compartment $i$, $r_{iS} = N_{iS}/C_{iS}$ | g N g C$^{-1}$ |
| $t$ | Time | d |
| Rates | | |
| $D_i$ | Decomposition of organic matter in compartment $i$ | g C kg$^{-1}$ d$^{-1}$ |
| $M_i$ | Mortality of microbes associated with compartment $i$ | g C kg$^{-1}$ d$^{-1}$ |
| $\phi_i$ | Net N mineralization by microbes associated with compartment $i$ | g N kg$^{-1}$ d$^{-1}$ |
| $f_C$ | Fraction of C transferred from residues + POM to MAOM via in vivo pathway | - |
| $f_N$ | Fraction of N transferred from residues + POM to MAOM via in vivo pathway | - |
| Parameters | | |
| $a$ | Parameter group, $a = e_P(1 - l)(1 - m)$ | - |
| $b$ | Insoluble fraction of the added residues (the fraction $1 - b$ is stabilized as MAOM at time zero) | - |
| $e$ | C-use efficiency of all microorganisms | - |
| $e_i$ | C-use efficiency of microorganisms in compartment $i$ | - |
| $l$ | Fraction of depolymerization products transferred from residues + POM to MAOM through the ex vivo pathway | - |
| $m$ | Fraction of necromass transferred from residues + POM to MAOM through the in vivo pathway | - |
| $r_B$ | N:C ratio of all microbial biomass | g N g C$^{-1}$ |
| $r_{iB}$ | N:C ratio of microbial biomass in compartment $i$ | g N g C$^{-1}$ |
| $\kappa$ | Proportionality coefficient, $\kappa = D_M/C_M \, (D_P/C_P)^{-1}$ | - |

### 2.1.2. Carbon mass balance equations

The C mass balance equations for substrates ($C_{PS}$) and microbial biomass ($C_{PB}$) in residues + POM are written as,

$$\frac{dC_{PS}}{dt} = - \underbrace{(1-l)D_P}_{uptake} - \underbrace{lD_P}_{ex\ vivo} + \underbrace{(1-m)M_P}_{recycled\ mortality} \,, \tag{1}$$

$$\frac{dC_{PB}}{dt} = \underbrace{(1-l)e_P D_P}_{growth} - \underbrace{M_P}_{mortality} \,, \tag{2}$$

where $D_P$ is the residues + POM decomposition rate, $M_P$ is the mortality rate, $l$ is the fraction of depolymerization products transferred to MAOM through the ex vivo pathway, $m$ is the fraction of necromass transferred to MAOM through the in vivo pathway (1-$m$ is the fraction recycled within the residues + POM compartment), and $e_P$ is the microbial C use efficiency (CUE).

Assuming that microbial biomass attains a quasi-equilibrium ($dC_{PB}/dt \approx 0$), so that growth equals mortality (i.e., $M_P \approx (1-l)e_P D_P$), and summing up substrate and microbial biomass, we can write a single equation for the total C in the residues + POM compartment ($C_P$),

$$\frac{dC_P}{dt} = \frac{d(C_{PS}+C_{PB})}{dt} = - \underbrace{(1-l)(1-e_P)D_P}_{respiration} - \underbrace{lD_P}_{ex\ vivo} - \underbrace{(1-l)me_P D_P}_{in\ vivo}, \tag{3}$$

with initial condition $C_P(0) = bC_{P,0}$, where $b$ is the insoluble residue fraction, which is retained as POM. Low values of $b$ represent residues with a soluble fraction that is mostly stabilized as MAOM without undergoing enzymatic reaction (Eq. (7)). If $b = 1$, no C is immediately stabilized, so that the initial condition for C in the residues + POM compartment is $C_P(1) = C_{P,0}$. For conciseness, we refer to residues with $b < 1$ as 'soluble' and to residues with $b = 1$ as 'insoluble' even though all residue types are at least partly soluble, but when $b = 1$ the soluble fraction is entirely used by microorganisms in the residues + POM compartment. The soluble fraction 1-$b$ is immediately transferred to MAOM, where it can be adsorbed or assimilated by microorganisms in that compartment.

Defining the parameter group $a = e_P(1-l)(1-m)$, Eq. (3) can be simplified to,

$$\frac{dC_P}{dt} = (a-1)D_P. \tag{4}$$

The C mass balance equations for substrate ($C_{MS}$) and microbial biomass ($C_{MB}$) in MAOM are written as,

$$\frac{dC_{MS}}{dt} = \underbrace{lD_P}_{ex\ vivo} + \underbrace{(1-l)me_P D_P}_{in\ vivo} - \underbrace{D_M}_{uptake} + \underbrace{M_M}_{mortality} \,, \tag{5}$$

$$\frac{dC_{MB}}{dt} = \underbrace{e_M D_M}_{growth} - \underbrace{M_M}_{mortality} \,, \tag{6}$$

where we adopted the same notation as for Eq. (1) and (2), except that now quantities refer to the MAOM compartment, as indicated by subscript $M$. The first two terms of Eq. (5) represent the C flows from the residues + POM compartment. We also assumed that all necromass produced by microbes in the MAOM compartment is recycled back into MAOM. Applying as

before the quasi-equilibrium approximation for microbial biomass ($dC_{MB}/dt \approx 0$), we determine the mortality rate (i.e., $M_M \approx e_M D_M$) and finally obtain a single equation for the total C in MAOM ($C_M$),

$$\frac{dC_M}{dt} = \frac{d(C_{MS}+C_{MB})}{dt} = \underbrace{lD_P}_{ex\ vivo} + \underbrace{(1-l)me_P D_P}_{in\ vivo} - \underbrace{(1-e_M)D_M}_{respiration}, \tag{7}$$

with initial condition $C_M(0) = (1-b)C_{P,0}$, where $1-b$ is the fraction of residue C immediately incorporated in MAOM. For insoluble residues with $b = 1$, the initial condition for C in the MAOM compartment is $C_M(0) = 0$.

Before proceeding, it is convenient to express the decomposition rate of MAOM as a function of the decomposition rate of residues and POM. One could argue that the kinetic constants for these two rates should be broadly correlated as they both respond to environmental conditions in similar ways (although POM can have slightly higher temperature sensitivity, Karhu

et al., 2019), but that MAOM decomposes more slowly than POM. Moreover, it is reasonable to expect that both decomposition rates scale approximately linearly with the C contents of the respective compartments (a reasonable approximation when considering long-term dynamics). This means that, as a first approximation,

$$\frac{D_M}{C_M} \approx \kappa\frac{D_P}{C_P} \rightarrow D_M \approx \kappa D_P \frac{C_M}{C_P}, \tag{8}$$

where $\kappa$ is the coefficient of proportionality between the (first order) kinetics constants of the decomposition rates. Values of $\kappa$ lower than one indicate that MAOM is decomposed slower than POM (as discussed in Section 2.2.2, $\kappa \approx 0.05$). This

assumption only implies a proportionality between the decay constants, while the actual rates will still be different depending on the relative abundance of C in residues + POM and MAOM.

### 2.1.3. Solution of the carbon mass balance equations in phase space

Equations (4) and (7) can be solved through time after specifying how the rates $D_P$ and $D_M$ vary with the state variables $C_P$ and $C_M$, as well as with environmental conditions. To remove part of the variability induced by environmental conditions and

185 to limit the number of model parameters, we move from this representation in the time domain to one in the phase space. To this aim, we now combine Eq. (4) and (7) to obtain a single ordinary differential equation with $C_P$ as independent variable and $C_M$ as dependent variable. This can be done by dividing Eq. (7) by Eq. (3) and simplifying $D_P$,

$$\frac{dC_M}{dt}\left(\frac{dC_P}{dt}\right)^{-1} = \frac{dC_M}{dC_P} = \frac{l+(1-l)me_P-(1-e_M)\kappa\frac{C_M}{C_P}}{a-1}. \tag{9}$$

The boundary condition for this equation is $C_M(bC_{P,0}) = (1-b)C_{P,0}$. This condition indicates that at the beginning of decomposition, the insoluble fraction ($b$) of added residues ($C_{P,0}$) is in the residues + POM compartment, while the soluble

fraction (1-$b$) is transferred to MAOM.

Eq. (9) is independent of the specifics of the kinetics laws used to describe decomposition rates and thus it is largely independent of time per se. However, Eq. (9) depends on the parameters regulating the two pathways of organic matter stabilization ($l$, $m$), the CUE of the two microbial groups ($e_P$, $e_M$), and the proportionality coefficient between the decomposition rates of MAOM and POM ($\kappa$).

To solve Eq. (9) and find the analytical relation $C_M(C_P)$, it is convenient to first normalize the C contents by the amount of added residue C—i.e., $c_P = C_P/C_{P,0}$ and $c_M = C_M/C_{P,0}$. This normalization allows comparing different datasets more easily, as all measured quantities are rescaled between 0 and 1, with values decreasing through time as decomposition progresses, until all the initially added residues ($c_{P,0} = 1$) are mineralized ($c_P = c_M = 0$). Moreover, the equations expressed in normalized form are independent of the units used to quantify inputs and mass in each compartment, and if needed it is easy to convert the normalized variables into absolute quantities by multiplying by the mass of added residue C. After normalizing, Eq. (9) becomes,

$$\frac{dc_M}{dc_P} = \frac{l + (1-l)me_P - (1-e_M)\kappa\frac{c_M}{c_P}}{a-1},$$ (10)

with boundary condition $c_M(b) = 1 - b$.

Eq. (10) is a non-autonomous ordinary differential equation with a compact analytical solution when $b = 1$ (insoluble residues),

$$c_M(c_P) = \left[ c_P - c_P^{\frac{\kappa(1-e_M)}{1-a}} \right] \frac{l+(1-l)me_P}{\kappa(1-e_M)+a-1}.$$ (11)

The full solution for the general case of partly soluble residues ($b < 1$) is reported in the Supplementary Materials (Section S2).

### 2.1.4. Nitrogen mass balance equations

Following the same rationale as for the C mass balance equations, we consider N in substrates ($N_{PS}$) and microbial biomass ($N_{PB}$) of residues + POM, as well as in substrates ($N_{MS}$) and microbial biomass ($N_{MB}$) of MAOM,

$$\frac{dN_{PS}}{dt} = -\underbrace{(1-l)D_P\frac{N_{PS}}{C_{PS}}}_{uptake} - \underbrace{lD_P\frac{N_{PS}}{C_{PS}}}_{ex\ vivo} + \underbrace{(1-m)M_P r_{PB}}_{recycled\ mortality},$$ (12)

$$\frac{dN_{PB}}{dt} = \underbrace{(1-l)D_P\frac{N_{PS}}{C_{PS}}}_{uptake} - \underbrace{M_P r_{PB}}_{mortality} - \underbrace{\phi_P}_{N\ mineralization},$$ (13)

$$\frac{dN_{MS}}{dt} = \underbrace{lD_P\frac{N_{PS}}{C_{PS}}}_{ex\ vivo} + \underbrace{mM_P r_{PB}}_{in\ vivo} - \underbrace{D_M\frac{N_{MS}}{C_{MS}}}_{uptake} + \underbrace{M_M r_{MB}}_{mortality},$$ (14)

$$\frac{dN_{MB}}{dt} = \underbrace{D_M\frac{N_{MS}}{C_{MS}}}_{uptake} - \underbrace{M_M r_{MB}}_{mortality} - \underbrace{\phi_M}_{N\ mineralization},$$ (15)

where the N:C ratios of residues + POM ($N_{PS}/C_{PS}$), MAOM ($N_{MS}/C_{MS}$), microbial biomass associated with residues + POM ($r_{PB} = N_{PB}/C_{PB}$), and microbial biomass associated with MAOM ($r_{MB} = N_{MB}/C_{MB}$) are used to convert C flow rates to N flow rates; and $\phi_P$ and $\phi_P$ are the net N mineralization rates of the two microbial groups. The net N mineralization rates are set so that the microbial N:C ratios are stable through time (Manzoni and Porporato, 2009); i.e., they are calculated as the differences between N demand for growth and N supply through uptake of organic N of the two respective microbial groups,

$$\phi_P = \underbrace{(1-l)D_P \frac{N_{PS}}{C_{PS}}}_{uptake} - \underbrace{(1-l)e_P D_P r_{PB}}_{growth\ demand} = (1-l)D_P\left(\frac{N_{PS}}{C_{PS}} - e_P r_{PB}\right), \tag{16}$$

$$\phi_M = \underbrace{D_M \frac{N_{MS}}{C_{MS}}}_{uptake} - \underbrace{e_M D_M r_{MB}}_{growth\ demand} = D_M\left(\frac{N_{MS}}{C_{MS}} - e_M r_{MB}\right). \tag{17}$$

These formulations for net N mineralization allow capturing both net N release if substrates are sufficiently rich in N ($N_{PS}/C_{PS} > e_P r_{PB}$, $N_{MS}/C_{MS} > e_M r_{MB}$) and net N immobilization when they cannot provide enough N for microorganisms ($N_{PS}/C_{PS} < e_P r_{PB}$, $N_{MS}/C_{MS} < e_M r_{MB}$). Recalling that the mortality rates can be expressed as a function of the decomposition rates thanks to the quasi-equilibrium approximation ($M_{PB} \approx (1-l)e_P D_P$ and $M_{MB} \approx e_M D_M$), we can now sum up substrate and microbial biomass and write the N mass balances for the total N in the residues + POM ($N_P$) and in the MAOM compartments ($N_M$),

$$\frac{dN_P}{dt} = \frac{d(N_{PS}+N_{PB})}{dt} = \underbrace{-lD_P \frac{N_P}{C_P}}_{ex\ vivo} - \underbrace{(1-l)me_P D_P r_{PB}}_{in\ vivo} - \underbrace{\phi_P}_{N\ mineralization}, \tag{18}$$

$$\frac{dN_M}{dt} = \frac{d(N_{MS}+N_{MB})}{dt} = \underbrace{lD_P \frac{N_P}{C_P}}_{ex\ vivo} + \underbrace{(1-l)me_P D_P r_{PB}}_{in\ vivo} - \underbrace{\phi_M}_{N\ mineralization}. \tag{19}$$

In these equations, we made the additional approximations $N_{PS}/C_{PS} \approx N_P/C_P$ and $N_{MS}/C_{MS} \approx N_M/C_M$, which are justified because the microbial biomass C and N contents are about two orders of magnitude smaller than the substrate C and N contents, respectively (Xu et al., 2013).

Substituting the definitions for the N mineralization rates from Eq. (16) and (17), we obtain more compact equations,

$$\frac{dN_P}{dt} = -D_P\left[\frac{N_P}{C_P} - (1-l)(1-m)e_P r_{PB}\right] = -D_P\left(\frac{N_P}{C_P} - a r_{PB}\right), \tag{20}$$

$$\frac{dN_M}{dt} = D_P\left[l\frac{N_P}{C_P} + (1-l)me_P r_{PB}\right] - D_M\left(\frac{N_M}{C_M} - e_M r_{MB}\right). \tag{21}$$

### 2.1.5.  Solution of the nitrogen mass balance equations in phase space

As for the C mass balance equations, we now combine Eq. (3), (20), and (21), and group parameters as before in $a = e_P(1-l)(1-m)$, to obtain two ordinary differential equations with $C_P$ as independent variable and $N_P$ and $N_M$ as dependent variables,

$$\frac{dN_P}{dt}\left(\frac{dC_P}{dt}\right)^{-1} = \frac{dN_P}{dC_P} = \frac{\frac{N_P}{C_P} - a r_{PB}}{1-a}, \tag{22}$$
$$N_P(bC_{P,0}) = bN_{P,0},$$

$$\frac{dN_M}{dt}\left(\frac{dC_P}{dt}\right)^{-1} = \frac{dN_M}{dC_P} = \frac{\kappa\frac{C_M}{C_P}\left(\frac{N_M}{C_M} - e_M r_{MB}\right) - l\frac{N_P}{C_P} - (1-l)me_P r_{PB}}{1-a}, \tag{23}$$
$$N_M(bC_{P,0}) = (1-b)N_{P,0}.$$

After normalizing $N_P$ by the amount of added residue N ($n_P = N_P/N_{P,0}$) and some algebraic manipulations, Eq. (22) becomes,

$$\frac{dn_P}{dc_P} = \frac{\frac{n_P}{c_P} - a\frac{r_{PB}}{r_0}}{1-a}, \quad n_P(b) = b, \tag{24}$$

where $r_0$ is the initial N:C ratio of the residues + POM ($r_0 = N_{P,0}/C_{P,0}$). If residues are insoluble ($b = 1$), Eq. (24) can be solved following Manzoni (2017) to obtain the N release curve,

$$n_P(c_P) = c_P \frac{r_{PB}}{r_0} + \left(1 - \frac{r_{PB}}{r_0}\right) c_P^{\frac{1}{1-a}}. \tag{25}$$

The general solution for $b < 1$ is reported in the Supplementary Information (Section S2). Eq. (25) reduces to a linear relation when $a \approx 0$ (i.e., if $l$ or $m$ are close to 1), $n_P(c_P) = c_P$. This property will be useful in the following.

Normalizing the N content in the MAOM compartment in Eq. (23) by the amount of added residue N ($n_M = N_M/N_{P,0}$) and after some algebraic manipulations we obtain,

$$\frac{dn_M}{dc_P} = \frac{\kappa\left(\frac{n_M}{c_P} - e_M\frac{r_{MB}}{r_0}\frac{c_M}{c_P}\right) - l\frac{n_P}{c_P} - (1-l)me_P\frac{r_{PB}}{r_0}}{1-a}, \quad n_M(b) = 1 - b. \tag{26}$$

Eq. (26) can be solved analytically thanks to the fact that $c_M$ and $n_P$ are known functions of $c_P$ (using Eq. (11) and (25), respectively). For simplicity, we now assume that the microorganisms associated with both substrate types have similar N:C ratio (i.e., $r_{PB} \approx r_{MB} = r_B$) and that residues are insoluble ($b = 1$). With these assumptions, the N release curve for MAOM is,

$$n_M(c_P) = \underbrace{c_P^{\frac{1}{1-a}}\left(1 - \frac{r_B}{r_0}\right)}_{=n_P(c_P) - c_P\frac{r_B}{r_0}}\left(c_P^{\frac{\kappa-1}{1-a}} - 1\right)\frac{l}{1-\kappa} + \underbrace{\left[c_P - c_P^{\frac{\kappa(1-e_M)}{1-a}}\right]\frac{l + (1-l)me_P}{\kappa(1-e_M)+a-1}\frac{r_B}{r_0}}_{=c_M(c_P)}, \tag{27}$$

where we highlighted how two of the terms on the right hand side of the equation are related to $n_P(c_P)$ (Eq. (25)) and $c_M(c_P)$ (Eq. (11)). The general solution for $b < 1$ is reported in the Supplementary Information (Section S2).

To summarize, Eq. (11), (25), and (27) constitute the solutions in phase space of the mass balance equations describing the dynamics of C and N in the residues + POM and MAOM compartments. These equations and their limiting cases under assumptions of only in vivo or only ex vivo stabilization are reported in Table 2. The shape of these equations depends on five parameters ($\kappa$, $l$, $m$, and microbial CUE and N:C ratio), which will be constrained using residues + POM and MAOM data, as described in Section 2.2.2.

**Table 2. Summary of analytical solutions of the dynamic model in phase space (C and N fractions are expressed as a function of the C fraction in residues + POM, $c_P$), including model variants parameterized to describe scenarios in which the in vivo or ex vivo stabilization pathways are dominant. The solutions reported here are derived from Eq. (11), (25), and (27) (insoluble residues, $b = 1$) by assuming for simplicity that all microbial groups have the same N:C ratio, $r_B$, and the same carbon use efficiency, $e$. The equations for $n_M(c_P)$ are written in a compact form as a function of $n_P(c_P)$ and $c_M(c_P)$. Parameter group $a$ is defined as $a = e(1-l)(1-m)$.**

| Scenario | C in MAOM, $c_M(c_P)$ | N in POM + residues, $n_P(c_P)$ | N in MAOM, $n_M(c_P)$ |
|---|---|---|---|
| General model | $\left[c_P - c_P^{\frac{\kappa(1-e)}{1-a}}\right]\frac{l+(1-l)me}{\kappa(1-e)+a-1}$ | $c_P\frac{r_B}{r_0} + \left(1-\frac{r_B}{r_0}\right)c_P^{\frac{1}{1-a}}$ | $\left(n_P - c_P\frac{r_B}{r_0}\right)\left(c_P^{\frac{\kappa-1}{1-a}}-1\right)\frac{l}{1-\kappa} + c_M\frac{r_B}{r_0}$ |
| Combined pathways: $l>0$, $m=1$, $a=0$ | $\left[c_P - c_P^{\kappa(1-e)}\right]\frac{l+(1-l)e}{\kappa(1-e)-1}$ | $c_P$ | $\left(c_P^\kappa - c_P\right)\left(1-\frac{r_B}{r_0}\right)\frac{l}{1-\kappa} + c_M\frac{r_B}{r_0}$ |
| Ex vivo: $l=1$, $a=0$ | $\left[c_P - c_P^{\kappa(1-e)}\right]\frac{1}{\kappa(1-e)-1}$ | $c_P$ | $\left(c_P^\kappa - c_P\right)\left(1-\frac{r_B}{r_0}\right)\frac{1}{1-\kappa} + c_M\frac{r_B}{r_0}$ |
| In vivo: $l=0$, $m=1$, $a=0$ | $\left[c_P - c_P^{\kappa(1-e)}\right]\frac{e}{\kappa(1-e)-1}$ | $c_P$ | $c_M\frac{r_B}{r_0}$ |

### 2.1.6. Contribution of the in vivo pathway to MAOM

Parameters $l$ and $m$ regulate how much C and N are transferred to MAOM, but the total amounts transferred throughout the whole decomposition process depends on these parameters, the residues + POM decomposition rate, and how much of the initial residue C and N is transferred immediately to MAOM. These total amounts are calculated by integrating through time the C and N flow rates from residues + POM to MAOM through the in vivo and ex vivo pathways,

$$\text{C to MAOM, in vivo pathway} = \int_0^\infty mM_P dt = \int_0^\infty m(1-l)e_P D_P dt, \tag{28}$$

$$\text{C to MAOM, ex vivo pathway} = (1-b)C_{P,0} + \int_0^\infty lD_P dt, \tag{29}$$

$$\text{N to MAOM, in vivo pathway} = \int_0^\infty mM_P r_{PB} dt = \int_0^\infty m(1-l)e_P D_P r_{PB} dt, \tag{30}$$

$$\text{N to MAOM, ex vivo pathway} = (1-b)N_{P,0} + \int_0^\infty lD_P r_{PS} dt. \tag{31}$$

In Eq. (29) and (31), the mass of residue C and N that is readily transferred to MAOM (i.e., the soluble fraction 1-$b$ of the added residues) is also accounted for in the calculation of the ex vivo contribution to MAOM.

The relative contribution of the in vivo pathway to MAOM ($f_C$ or $f_N$) can be then calculated as the ratio between the mass of C or N transferred from the microbial biomass in residues + POM to MAOM over the total mass of C or N transferred from POM to MAOM,

$$f_C = \frac{\int_0^\infty m(1-l)e_P D_P dt}{(1-b)C_{P,0} + \int_0^\infty [m(1-l)e_P D_P + lD_P]dt} \approx \frac{m(1-l)e_P}{m(1-l)e_P + l}, \tag{32}$$

$$f_N = \frac{\int_0^\infty m(1-l)e_P D_P r_{PB} dt}{(1-b)N_{P,0} + \int_0^\infty [m(1-l)e_P D_P r_{PB} + lD_P r_{PS}]dt} \approx \frac{m(1-l)e_P r_{PB}}{m(1-l)e_P r_{PB} + lr_0}, \tag{33}$$

where in the last equalities of both equations we assumed that the residues were insoluble ($b = 1$). In Eq. (33) we also approximated the time-varying N:C ratio of the residues + POM substrates ($r_{PS}$) with the time-invariant initial residue N:C ratio ($r_0$). This allows taking out from the integrals in Eq. (33) all coefficients and N:C ratios, so that the integrals can be simplified, as also done in Eq. (32). As demonstrated in Section 3.2, this approximation is supported by the data. Simpler formulas for $f_C$ and $f_N$ can be easily obtained for the different model variants (Table 3).

**Table 3. Summary of analytical formulas for the relative contributions of the in vivo pathway to MAOM C and N ($f_C$ and $f_N$ from Eq. (32) and (33), respectively). As for results reported in Table 2, we consider insoluble residues ($b = 1$) and assume for simplicity that all microbial groups have the same N:C ratio, $r_B$, and the same carbon use efficiency, $e$.**

| Scenario | $f_C$ | $f_N$ |
|---|---|---|
| General model | $\dfrac{m(1-l)e}{m(1-l)e+l}$ | $\dfrac{m(1-l)er_B}{m(1-l)er_B + lr_0}$ |
| Combined pathways: $l > 0$, $m = 1$, $a = 0$ | $\dfrac{(1-l)e}{(1-l)e+l}$ | $\dfrac{(1-l)er_B}{(1-l)er_B + lr_0}$ |
| Ex vivo: $l = 1$, $a = 0$ | 0 | 0 |
| In vivo: $l = 0$, $m = 1$, $a = 0$ | 1 | 1 |

## 2.2. Data and model parameterization

### 2.2.1. Data retrieval and processing

Residue-derived C and N contents in undecomposed residues, POM, and MAOM were collated from published studies (Table S1). In most studies, [13]C or [14]C was used as C tracer and [15]N as N tracer; in a few studies, residue-derived C and N were estimated by difference between residue-amended and control treatments. We considered plant or microbial residues, but not leachates or biochar. Residues were often (but not always) separated before soil fractionation as fragments larger than 2 mm. Finely ground residues were instead recovered as POM (in that case we report the sum of residues and POM C or N). POM was generally isolated via density fractionation (light fraction with density lower than 1.6 to 2 g cm[-3]) or size fractionation (coarse fraction with size larger than 53 μm). Where both free and occluded POM were reported, they were combined into a single POM fraction. MAOM was generally defined as heavy fraction (density higher than 1.6 to 2 g cm[-3]) or fine fraction (size smaller than 53 μm). Published data was obtained from tables and digitized figures or was provided by the authors. In some cases, authors provided additional unpublished data to complete the datasets.

Data sources were selected to guarantee some degree of comparability across studies. Studies where residue C or N were traced in aggregates, but where it was not possible to distinguish between POM and MAOM within aggregates were not considered.

Reported negative values for any of the considered quantities were removed, but if primary data showed major inconsistencies (e.g., negative fractions of remaining residues) that could not be explained even after contacting the authors, the whole study was excluded. After this screening, the database contained data from 42 published articles (Almeida et al., 2021; Antonio Telles Rodrigues et al., 2022; Buckeridge et al., 2022; Canisares et al., 2023; Cheng et al., 2023; Cotrufo et al., in review, 2015, 2022; Craig et al., 2022; Dai et al., 2022; Duan et al., 2023; Even and Cotrufo, 2024; Fang et al., 2019; Ferreira et al.,

2021; Fulton-Smith and Cotrufo, 2019; Haddix et al., 2020, 2016; Huys et al., 2022a; Kölbl et al., 2006, 2007; Kou et al., 2023b; Lavallee et al., 2018; Leichty et al., 2021; Lian et al., 2016; Liang et al., 2023; Liebmann et al., 2020; Lyu et al., 2023; Magid et al., 2002; Mitchell et al., 2018; Neupane et al., 2023; Nunez et al., 2022; Nyamasoka-Magonziwa et al., 2022; Oliveira et al., 2021; Poeplau et al., 2023; Pries et al., 2017, 2018; Ridgeway et al., 2022, 2023b; Schiedung et al., 2023; Sokol et al., 2019; Su et al., 2020; Throckmorton et al., 2015; Wang et al., 2017; Witzgall et al., 2021; Xu et al., 2022). Some datasets were

directly accessible from online repositories (Buckeridge, 2021; Craig et al., 2021; Huys et al., 2022b; Kou et al., 2023a; Ridgeway et al., 2023a). Several of the datasets within this database are incomplete because, depending on the specific experimental design, only C vs. C and N contents, or only POM vs. combined residues and POM had been measured. Due to these gaps, 36 out of the total 42 datasets are used in the following analyses.

Minor data processing and harmonization were also performed. In the few studies reporting values of replicate measurements,

replicates in each treatment and date were averaged. When the sum of residue-derived C in POM and MAOM did not match the amount of residue-derived C in the bulk soil (typical for physical fractionation, given mass and C recoveries may vary from 100%), we recalculated the fractions of residue-derived C in POM and MAOM as products of the fraction of residue-derived C in bulk soil times the relative contribution of C in POM versus the sum of C in POM and MAOM, and similarly for C in MAOM.

If not reported, the C content of the residues (g C per g of residue dry weight) was assumed equal to that of species in the same family that was provided in other studies of the database. Fractions of remaining residue C were approximated by the fractions of remaining residue dry mass if C contents were not reported for all measurement times. Finally, all C and N contents were normalized by the residue C and N contents added to the soil samples. In this way, C and N in undecomposed residues, POM and MAOM were all expressed as fractions of remaining residue C and N (as in the model equations).

In addition to residue-derived C and N in the soil fractions, we also collected from the original data sources information on residue and soil properties, as well as climatic conditions at the sampled sites, including initial residue C:N ratio, soil texture and total organic C content, and temperature during the laboratory or field incubation. If detailed texture data was not reported, percentages of sand, silt and clay were inferred from the soil texture qualitative description provided in the data source. If no specific value of mean temperature during the field incubation was reported, we used the mean annual temperature at the

incubation site. Generally, incubations in the field lasted more than one year, making the mean annual temperature representative of actual incubation conditions.

### 2.2.2. Model parameter estimation

The model was fitted to residue-derived C and N contents in both residues + POM and MAOM fractions. The number of free parameters was reduced by assuming that both microbial groups have the same CUE ($e = e_P = e_M$) and N:C ratio ($r_B = r_{PB} = r_{MB}$). The latter parameter was assumed fixed $r_B = 0.13$ g N g C$^{-1}$, corresponding to the global average microbial C:N ratio of 7.6 g C g N$^{-1}$ (Xu et al., 2013). With these assumptions, the model solutions $c_M(c_P)$, $n_P(c_P)$, and $n_M(c_P)$ (Table 2) have still four free parameters: $e$, $\kappa$, $l$, $m$. These parameters have partly similar effects, so fitting all of them could lead to equifinality issues, requiring us to constrain some of these parameters before fitting the others.

The relative decomposability $\kappa$ was estimated as 0.05, based on the decay constants used in the MEMS-2 model (Zhang et al., 2021). In MEMS-2, the ratios of the decay constants for decomposition of MAOM and hydrolysable residues, oxidizable residues, and POM are $\approx 2 \times 10^{-2}$, $\approx 5 \times 10^{-2}$, $\approx 10^{-1}$, respectively. The higher ratio $\kappa \approx 10^{-1}$ is also comparable to that estimated by Guo et al. (2022). Considering that here residues and POM are merged in a single compartment including also chemically recalcitrant compounds, we considered the intermediate value $\kappa = 0.05$. We also attempted to constrain other parameters and fit $\kappa$ to the data instead of setting a fixed value. However, this approach was unsuccessful, as fitting was poor for most datasets. In Section 3.2 we present arguments for constraining also the value of $m$, so that the remaining parameters $e$ and $l$ can be fitted to the $c_P$ and $c_M$ data using Eq. (11) that provides the relation between them $c_M(c_P)$ (a lower bound for $e$ values was set to 0.02). To this aim, we used all the time series with at least three $c_P$ and $c_M$ data pairs after grouping data from similar treatments, but not from different soils or treatments involving N additions, as those are expected to affect organic matter stabilization (38 time series in total). Too few datasets included residue-derived N contents in soil fractions for a systematic analysis, so model parameters were fitted only to the C data, except for a few examples. Model fitting was performed by minimizing the square errors between measurements and data, using the function lsqcurvefit in Matlab (MathWorks, 2018).

### 2.2.3. Statistical analysis of model results

The estimated model parameters were predicted using as independent variables: C:N ratio of the added residues, clay content, soil organic C content (as an index of overall C availability), and incubation temperature (laboratory temperature or air temperature at the field site where the litter was incubated). The data were fitted with a linear mixed effect model including interactions of clay content with residue C:N ratio and soil organic C content, and with data source as a random factor, using the function fitlme in Matlab (MathWorks, 2018).

## 3. Results

### 3.1. Model behavior

In general, the data show that C and N from the residues + POM compartment accumulate in MAOM in the early decomposition phase, while later both residues + POM and MAOM compartments lose mass (Appendix A). The same general trend is captured by modeled phase-space trajectories, but these trajectories are modulated by the dominant stabilization pathways and other model parameters (Fig. 2). For all parameter values, decreasing $c_P$ during decomposition causes an initial increase in both $c_M$ and $n_M$ because C and N are transferred from residues + POM to MAOM (top and bottom rows in Fig. 2). However, towards the end of the decomposition process, as $c_P$ nears zero, transfer to MAOM is lower than mineralization of MAOM, so that both $c_M$ and $n_M$ start decreasing to eventually also reach zero. Decomposition of residues + POM also causes $n_P$ to decrease, although in some scenarios N is preferentially retained in this pool before being transferred to MAOM or mineralized (downward concavity of the curves in the central row in Fig. 2).

Differences in the model behavior emerge when comparing predictions under contrasting stabilization pathways. If the in vivo pathway is dominant ($l = 0$, $m = 1$; blue), at a given $c_P$, less C accumulates in MAOM (lower $c_M$) compared to a scenario where the ex vivo pathway is dominant ($l = 1$, $m = 0$; orange). This lower C accumulation is due to respiration that removes C before necromass is formed and transferred to MAOM. In contrast, more N accumulates in MAOM (higher $n_M$) if the in vivo pathway is dominant. This higher N accumulation is due to the N-enriched necromass from the residues + POM compartment. Because in the in vivo scenario necromass is not recycled within the residues + POM compartment ($m = 1$), the C:N ratio of that compartment remains constant. Microorganisms might still need to immobilize N when feeding on N poor residues + POM (Eq. (16)), but the acquired N supports the production of biomass that is eventually transferred to MAOM. The mix scenario, with simultaneous in vivo and ex vivo stabilization (green curves), leads to trajectories of $c_M$ and $n_M$ that are intermediate between the two more extreme scenarios.

Neither the in vivo or the ex vivo pathway lead to preferential N retention in residues + POM (central row in Fig. 2). Mathematically, this pattern is explained by the fact that $n_P = c_P$ when either $l = 1$ or $m = 1$ (both resulting in $a = 0$). However, in the mix scenario, preferential N retention in the residues + POM compartment occurs, as indicated by the downward concavity of the green curves in the central row of Fig. 2.

Increasing the C:N ratio of the added residues (solid vs. dashed curves in the 2nd column of Fig. 2) causes relatively stronger N retention in residues + POM in the mix scenario, and relatively higher N accumulation in MAOM, due to the higher N immobilization needed to satisfy the microbial N demand in both compartments. The same mechanism also causes higher N retention when increasing microbial CUE (solid vs. dashed curves in the 3rd column of Fig. 2). However, higher CUE also increases C accumulation in MAOM (while residue C:N has no effect on $c_M$) because less C is lost through respiration when CUE is higher.

Finally, increasing the MAOM decay constant relative to the decay constant of residues + POM (higher $\kappa$) causes lower retention of C and N in MAOM, and more curvilinear trajectories as $c_P$ decreases (solid vs. dashed curves in the 4th column of

Fig. 2). This pattern differs from the nearly linear accumulation (and very late decomposition) of C and N in MAOM when $\kappa$ is low.

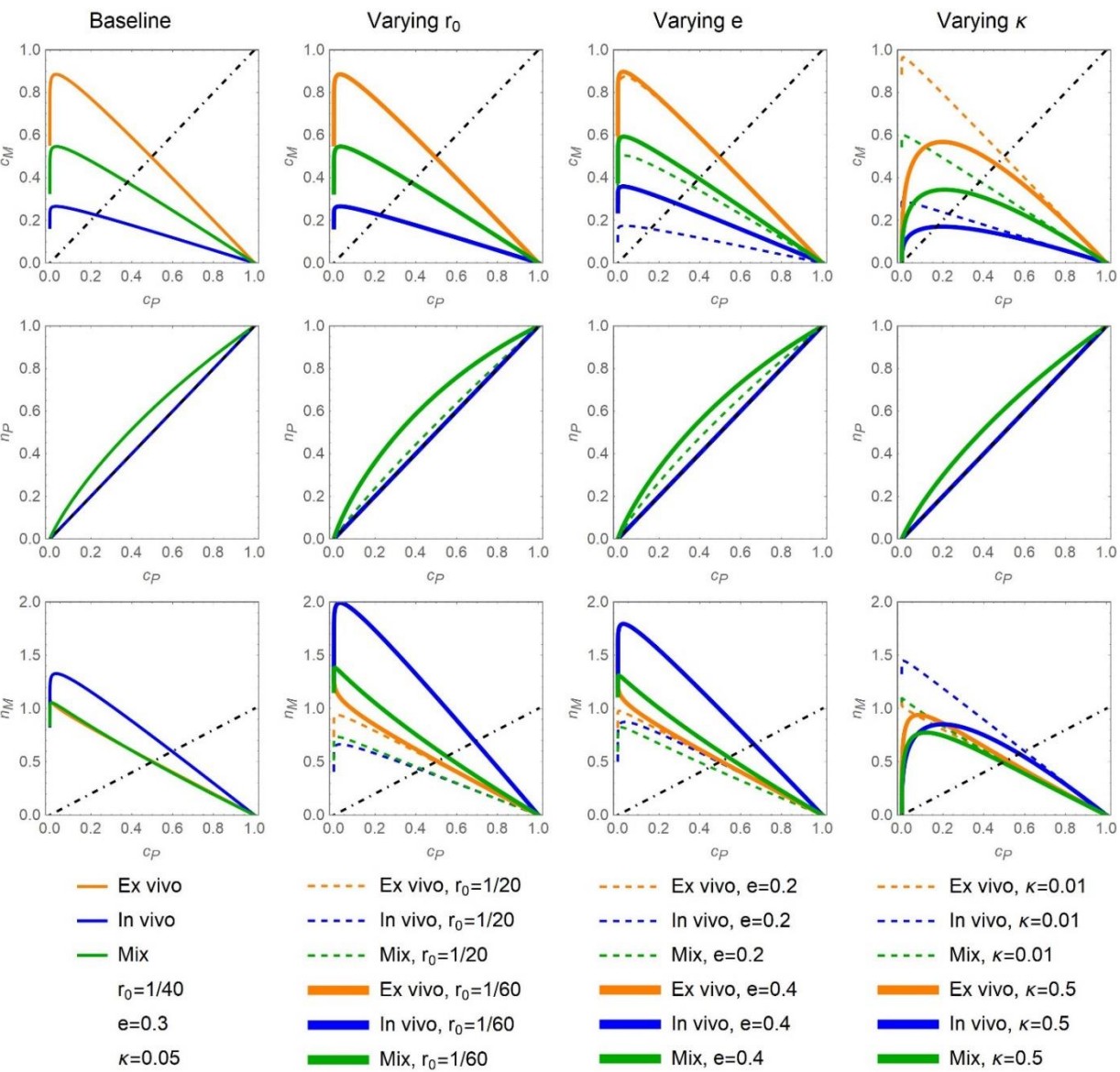

Figure 2. Fraction of added C in MAOM, $c_M$ (top row), fraction of added N in residues + POM, $n_P$ (center row), and fraction of added N in MAOM, $n_M$ (bottom row), as a function of the fraction of added C in residues + POM, $c_P$, under different stabilization pathway scenarios (colors) and when varying the values of model parameters around baseline values shown in the 1st column: residue N:C ratio, $r_0$ (2nd column), microbial carbon use efficiency, $e$ (3rd column), and ratio between the decay constants of MAOM and residues + POM decomposition, $\kappa$ (4th column). Three stabilization scenarios are considered: dominant ex vivo stabilization ($l = 1$; orange), dominant in vivo stabilization ($l = 0, m = 1$; blue), and a combination of pathways denoted by 'mix' ($l = 1/2, m = 1/2$; green). In all panels, residue decomposition progresses from right to left along the curves, as $c_P$ decreases. The dot-dashed black lines indicate 1:1 lines, which represent equality between the fractions of added C or N shown on the y-axes and $c_P$ shown on the x-axes; the added residues are assumed to be insoluble ($b = 1$).

## 3.2. Stabilization pathways – mathematical analysis

Two lines of evidence help us constrain parameters $l$ and $m$, which represent the MAOM stabilization pathways. First, during decomposition, the C:N ratio of the combined residues + POM compartment remains similar to the initial residue C:N (Fig. 3A). In general, POM is expected to have lower C:N than the residues, because necromass recycling enriches the decomposing residues in N. Therefore, the observed stable C:N in the residues + POM compartment is surprising. Stable C:N implies that either microbial necromass recycling is low in the residues + POM compartment or all depolymerization products are transferred to MAOM so biomass growth is low. The first explanation corresponds to C and N in necromass being stabilized through the in vivo pathway, which in our mathematical framework implies $m \approx 1$. The second explanation requires instead that most C and N released during residues + POM decomposition is transferred to MAOM through the ex vivo pathway, corresponding to $l \approx 1$.

Mathematically, stable C:N in the residues + POM compartment requires $a \approx 0$, so that $n_P(c_P) \approx c_P$ (Eq. (25)), or—after converting variables back to actual C and N contents—$N_P(C_P) \approx r_0 C_P$. Fitting Eq. (25) to all $n_P$ and $c_P$ pairs in the dataset we found $a = 0.012$ when considering the median $r_0$, confirming that the C:N ratio of the residues + POM compartment is nearly constant. The parameter group $a$ depends on both $l$ and $m$, and $a$ is approximately zero when either $l \approx 1$ (ex vivo pathway) or $m \approx 1$ (in vivo pathway). Therefore, this first argument points to one of the alternative scenarios for the model parameterization: either $l \approx 1$ (in such a case the value of $m$ is inconsequential) or $m \approx 1$ (with $l$ still to be determined). It is also possible that $a \approx 0$ due to simultaneously low value of $e$ and high values of $l$ and $m$, but microbial carbon use efficiency in incubation studies with high organic matter availability is likely in the range 0.1 to 0.3 at least in the early phase of decomposition (e.g., CUE values reported for one of the incubation studies, Craig et al., 2022). We thus discard this third possibility and focus on the alternatives $l \approx 1$ or $m \approx 1$.

The second line of evidence points to significant release of C through respiration as C is transferred from residues + POM to MAOM (Fig. 3B). It is likely that part of the depolymerization products are already metabolized by microbes in residues + POM (or even via extracellular oxidative metabolism, Maire et al., 2013) with release of $CO_2$. Mathematically, we can quantify the rate of change in $c_M$ as $c_P$ decreases in the early phase of decomposition—i.e., we can calculate from Eq. (11) $dc_M/dc_P$ for $c_P \to 1$ and $c_M \to 0$,

$$\left. \frac{dc_M}{dc_P} \right|_{c_P \to 1} = -\frac{l+(1-l)me}{1-e(1-l)(1-m)}. \tag{34}$$

This derivative is always negative, because decreasing $c_P$ causes an increase in $c_M$, but the specific values depend on the parameter choice. In the ex vivo scenario, $l \approx 1$ and $dc_M/dc_P \approx -1$ indicating no C loss during the transfer from POM + residue to MAOM. This is a clearly unrealistic scenario because data suggest significant C loss. In fact, the measured $c_M$ are lower than $1 - c_P$ (dashed line in Fig. 3B), indicating that not all C from residues + POM is transferred to MAOM.

In contrast, in the in vivo scenario, $m \approx 1$ and $dc_M/dc_P \approx -l - (1-l)e$. The largest—but still reasonable—increase in MAOM as residues + POM is decomposed can be quantified from $c_M$ and $c_P$ data through the upper quartile boundary line

shown in red in Fig. 3A. The slope of this line is $dc_M/dc_P = -0.37$. This value corresponds to a reasonable $e = 0.37$ if $l = 1$ or to any combination of $l$ and $e$ satisfying $0.37 = l + (1-l)e$. For $l$ to be larger than zero, $e < 0.37$. For $e$ in the range 0.1 to 0.3, we find $l$ between 0.1 and 0.3.

To summarize these initial results based on a simple mathematical analysis of the model combined with measured C and N contents in soil fractions, we can narrow down the range of plausible parameter values to $m \approx 1$ and $l < 0.3$. In the following,

we will set $m = 1$, while conservatively letting $l$ free to vary. This allows us to determine through least square fitting of individual data time series the parameters regulating the stabilization pathway and thus the relative contribution of each pathway to C and N stabilization (Section 3.4).

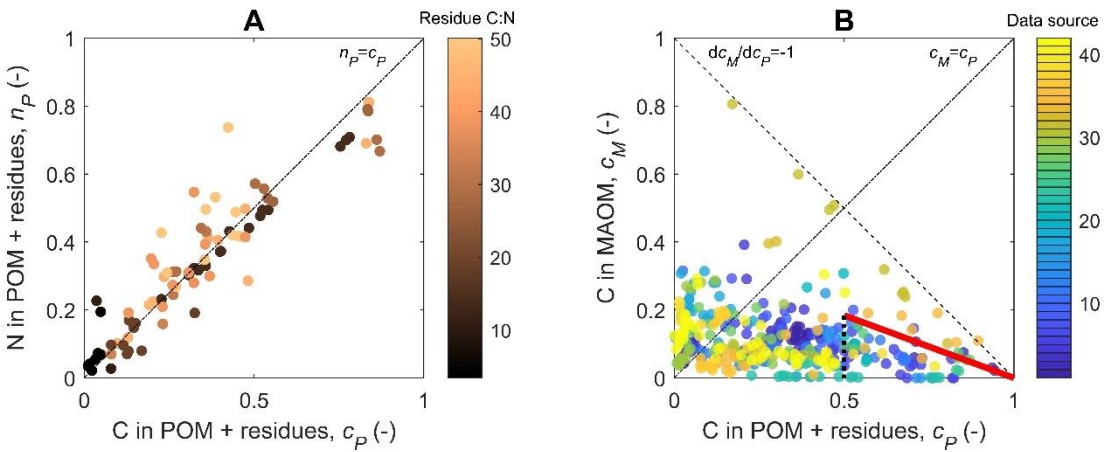

**Figure 3. A) Relation between the fractions of added residue N ($n_P$) and C ($c_P$) recovered in residue + particulate organic matter**
**(POM); lighter colors refer to residues with increasing C:N ratio. B) Relation between the fraction of added residue C recovered in mineral associated organic matter (MAOM), $c_M$, and $c_P$; different colors indicate different data sources. In both panels, residue decomposition progresses from right to left along the curves, as $c_P$ decreases. The dot-dashed black lines indicate 1:1 lines. In A, the 1:1 line corresponds to the equation $n_P(c_P) = c_P$ (i.e., $N_P(C_P) = r_0 C_P$), which is the solution of the model for both 'ex vivo' and 'in vivo' scenarios (Table 2). In B, the dashed black line indicates the trajectory of conversion from residues + POM to MAOM without**
**any C loss via respiration, whereas the thick red line is the upper quartile boundary line for the early decomposition phase ($0.5 \leq c_P \leq 1$). Data source numbers refer to the 'source ID' in the database (Manzoni et al., 2024).**

### 3.3. Examples of model calibration on individual time series

Parameters $e$ and $l$ were calibrated to datasets with at least three pairs of data points (Section 2.2.2). Examples of data time series and fitting of both $c_M(c_P)$ and $n_M(c_P)$ are shown in Fig. 4. In the first example (Fig. 4A, D, G), the residues (microbial necromass) were labile and N rich, so they were decomposed rapidly. As a result, sampling took place when most of the residues had already been decomposed ($c_P \approx 0$), so that both C and N in MAOM decrease. In the second example, representing the addition of a residue with intermediate C:N (Fig. 4B, E, H), C accumulates very slowly in MAOM as C in residues + POM is decomposed, whereas N in MAOM increases through time and as $c_P$ decreases. In the third example (Fig. 4C, F, I), relatively N poor residues exhibit strong N immobilization and accumulation of N in MAOM. In the last two examples, more N than C accumulate in MAOM at a given time or $c_P$ value (compare Fig. 4B and 4E or Fig. 4C and 4F), indicating preferential retention and stabilization of N when residues with high C:N are decomposed (as also shown in Figure 2). These examples show that data can be representative of early (last two examples) and later phases (first example) in the same stabilization pattern, which are linked through a single curve in the phase space. Therefore, datasets might appear inconsistent across studies ($c_M$ and $n_M$ increasing vs. decreasing through time), but the underlying dynamic behavior is the same. Despite similar underlying dynamics, the fitted parameters are different across studies, reflecting contrasting residue type (plant vs. microbial necromass), soil characteristics, and experimental conditions, as shown in the next section.

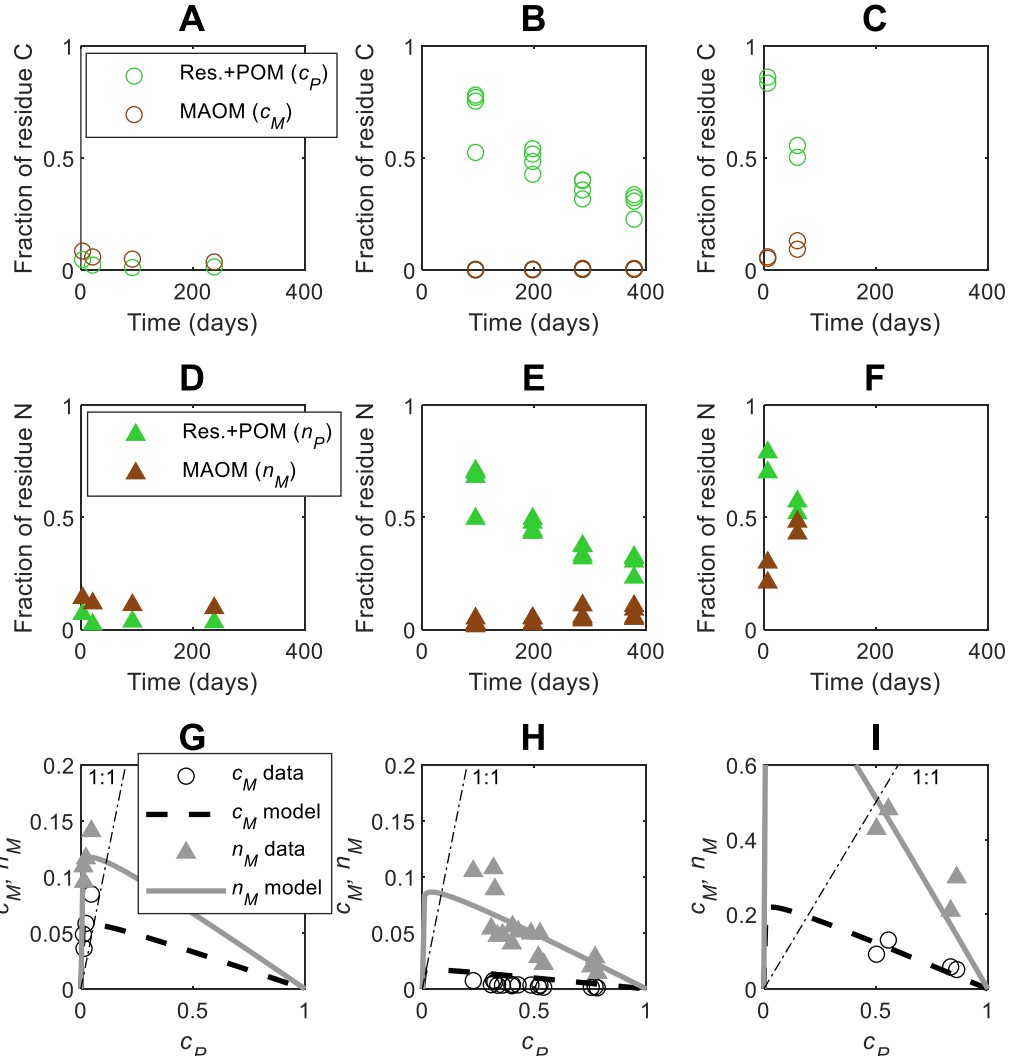

**Figure 4. Examples of data time series from incubations (A-F) and model fitting in the phase space (G-I), for residues with increasing residue C:N from the left to the right column: A-C) fractions of added C in residues + POM ($c_P$, green circles) and MAOM ($c_M$, brown circles) as a function of time; D-F) fractions of added N in POM + residues ($n_P$, green triangles) and MAOM ($n_M$, brown triangles) as a function of time; G-I) $c_M$ (black circles) and $n_M$ (gray triangles) as a function of $c_P$. In G-I, we fitted parameters $e$ and $l$ in the functions $c_M(c_P)$ and $n_M(c_P)$ with $m = 1$ and $\kappa = 0.05$ (Table 2). Data are from: A, D, G) Buckeridge et al. (2022) (residues: *Escherichia coli* necromass, C:N=3.4), B, E, H) Mitchell et al. (2018) (residues: *Chloris gayana*, C:N=14.2), and C, F, I) Lavallee et al. (2018) (residues: *Andropogon gerardii*, silt soil, C:N=28.2). In G-I, residue decomposition progresses from right to left along the curves, as $c_P$ decreases; the dot-dashed lines represent equality between the fractions of added C or N shown on the y-axes and $c_P$ shown on the x-axes.**

### 3.4. Stabilization pathways – general patterns

We focus now on fitting of $c_M(c_P)$, as there are too few datasets including N in MAOM to draw general conclusions. The values of $e$ and $l$ obtained from fitting $c_M(c_P)$ were weakly correlated (Pearson correlation coefficient = 0.27) indicating that despite constraining other parameters, there remain mild equifinality issues with the two calibrated parameters. Values of $e$ were below ≈0.2 (Fig. 5), with the lowest values from datasets with minimal accumulation of C in MAOM (Mitchell et al., 2018). Values of $l$ were generally lower than 0.1, indicating that less than 10% of depolymerized C is transferred to MAOM

and confirming our expectations from the mathematical analysis (Section 3.2). Low values of $l$ might be associated with large depolymerization rates ($D_P$ in Fig. 1) and low microbial CUE, so that the actual rate of C transfer to MAOM via the ex vivo pathway could still be large ($f_C$ calculated from Eq. (32)), but that was not the case. Indeed, the median relative contribution of the in vivo pathway to MAOM formation is ≈75 % (Fig. 5), but with a large variability. Notably, the contribution of the in vivo pathway is larger for N ($f_N$ calculated from Eq. (33)), with a median value of 96%.

Next, we tested how the estimated parameters $e$ and $l$ are affected by residue C:N ratio, soil properties (clay fraction, SOC content), and incubation conditions (temperature). When accounting for the combined effects of all variables and grouping data by source with a linear mixed effect model, we found that $e$ decreased with increasing residue C:N (Fig. 6). The in vivo contributions $f_C$ and $f_N$ increased in more clayey soils, but the effect of clay was less positive in C rich soil and when adding N poor residues (significant negative interactions of clay fraction with SOC and residue C:N; Fig. 6).

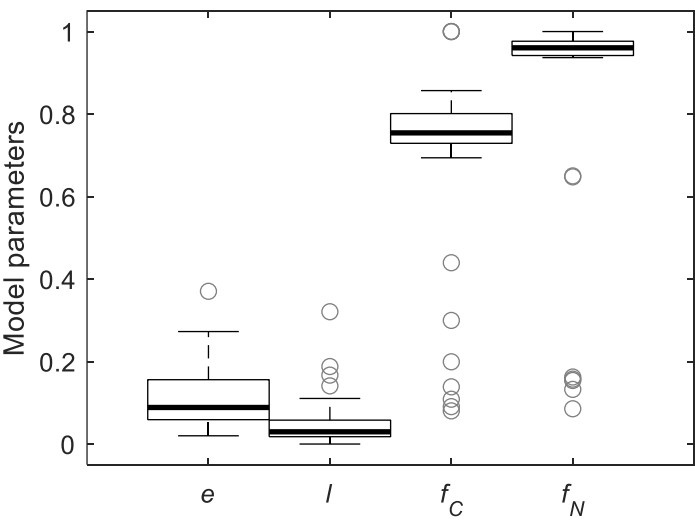

**Figure 5. Box plots of fitted model parameters and relative contributions of the in vivo pathways to MAOM C and N: microbial carbon use efficiency ($e$), fraction of depolymerized products transferred from POM + residue to MAOM ($l$), fractions of C and N transferred from POM to MAOM via in vivo pathway (respectively $f_C$ and $f_N$). Each box shows median and quartiles, and whiskers represent extreme values (1.5 times the interquartile range).**

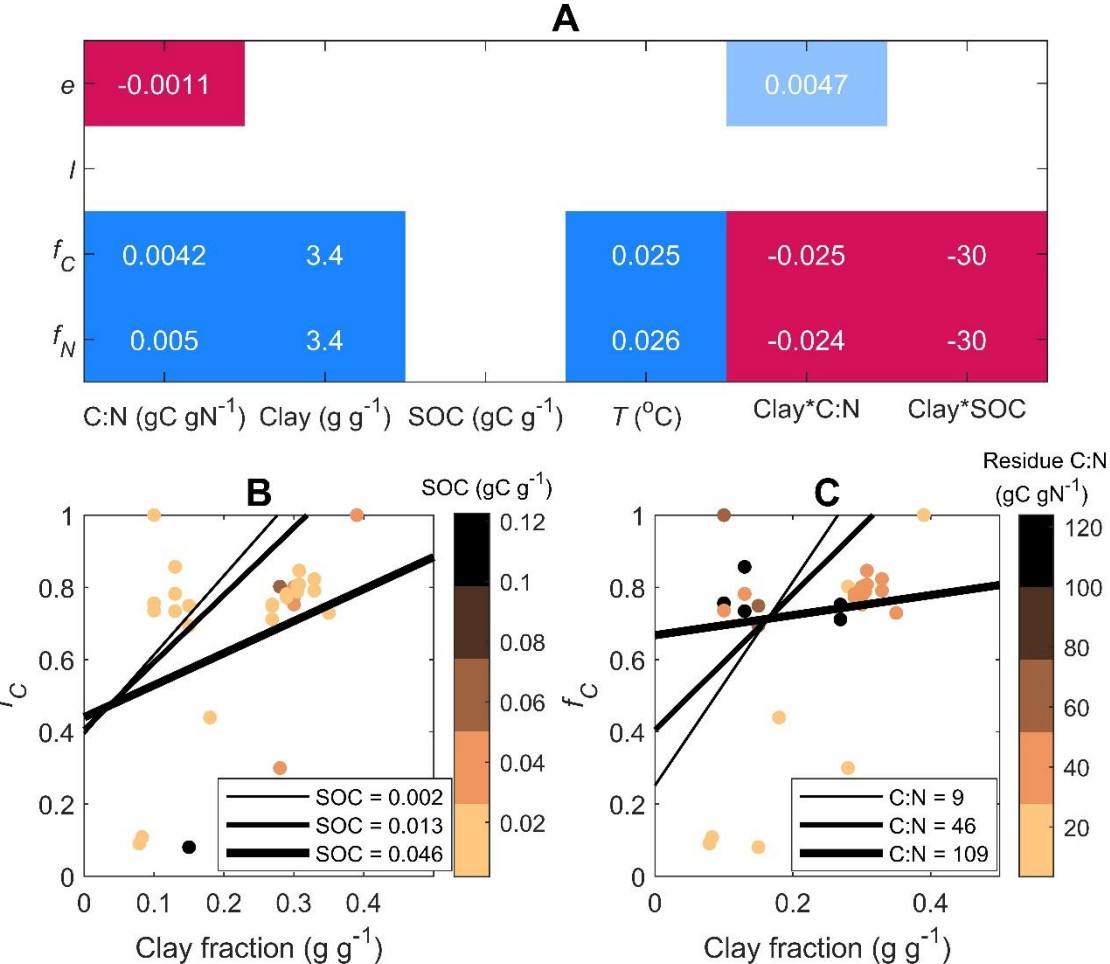

**Figure 6. Results of linear mixed effect models predicting model parameters (*e* and *l*) and relative contributions of the in vivo pathway to MAOM C and N ($f_C$ and $f_N$, respectively) as a function of residue C:N ratio (C:N), soil clay fraction, soil organic carbon content (SOC), incubation temperature (*T*), and the interactions of clay fraction with C:N and SOC, with data source as a random factor. A) Model coefficients: colors indicate the direction of the effect (red: negative, blue: positive) and shading intensity indicates the significance of the effect (blank: not significant, light colors: $0.05<p<0.1$, dark colors: $p<0.05$); marginal coefficients of determination: 0.89, 0.96, 0.65, and 0.80 for *e*, *l*, $f_C$, and $f_N$, respectively. The bottom panels show model predictions of $f_C$ as a function of clay fraction when varying B) SOC content (gC g⁻¹) and C) residue C:N ratio (gC gN⁻¹) as indicated by the thickness of the lines (values of SOC and C:N represent 10, 50, and 90th percentiles of the observations). Data points color coded by SOC (B) and residue C:N (C) are also shown.**

# 4. Discussion

## 4.1. Model design and solution in phase space

We considered only two compartments in our model, in contrast to other more complex C and N cycling models describing also dissolved organic matter, microbial biomass (which is here assumed to be in quasi equilibrium), occluded organic matter, and MAOM with different degrees of availability to decomposition (Abramoff et al., 2018; Guo et al., 2022; Zhang et al., 2021). However, models with more than two compartments might not fit POM and MAOM data better, while having worse equifinality issues (Guo et al., 2022). Therefore, our model design balances the need to both represent (at least in a simplified 510 way) the previously hypothesized stabilization pathways and minimize the number of parameters to fit.

Different from previous models, here we study the dynamics of one state variable (C in MAOM, N in residues + POM, or N in MAOM) as a function of another variable (C in residues + POM). This approach allows focusing on relations among variables rather than the temporal progression of the decomposition and stabilization process. This is particularly useful when the temporal trajectories are very different among datasets (e.g., Figure 4A-F), while in the phase space data start exhibiting 515 more consistent (and simpler) trends (Figure 4G-I). From a modeling perspective, using time series would require calibrating not only the parameters regulating the partitioning of C and N flows into different pathways, but also decay constants and those parameters that capture the effects of environmental conditions on the rate of decomposition—e.g., parameters in soil moisture or temperature rate modifiers (Bauer et al., 2008). Moreover, temporal dynamics depend on the chosen kinetics for decomposition, whereas our approach is largely independent of the kinetics (except for the assumption that the ratio of MAOM 520 and residues + POM decomposition rates scales approximately as the ratio of C contents in those two compartments).

While not aiming to model C and N stabilization, previous work described N release from decomposing residues following this approach, leading—albeit through different derivations—to an equation linking residue N to residue C that is formally equivalent to Eq. (25) (the theory was developed by Bosatta and Ågren, 1985; Manzoni et al., 2008). That equation was then fitted to measured fractions of remaining C and N in litterbag incubations, to estimate the CUE of residue decomposers (Bosatta 525 and Ågren, 1985; Manzoni, 2017; Manzoni et al., 2008) or their threshold element ratio (i.e., the C:N ratio below which net N mineralization starts) (Ågren et al., 2013). In our application, we use the same equation to infer the constraint $m = 1$, but we estimate the parameter e (conceptually analogous to CUE) with the analytical equation linking C in MAOM to C in residues + POM.

Moreover, our approach allows finding analytical solutions that provide mathematical insights into these processes. Besides 530 the already mentioned application of the N vs. C relation to constrain parameter *m*, the analytical relation between MAOM C and residues + POM C (i.e., $c_M(c_P)$) allowed determining limit values for parameter *l* by studying the slope of the $c_M(c_P)$ function at the beginning of decomposition. These insights would not be possible when solving numerically a more complex model.

## 4.2. Model limitations

Our model was designed to match the type of data available—residue, POM, and MAOM fractions measured at coarse temporal resolution from soils sampled from different ecosystems and land uses. We also aimed for full analytical tractability. These two requirements set constraints on the model complexity and the number of parameters that could be estimated from the data. These constraints in turn imply the following simplifications and approximations that might limit the model applicability:

-    Model structure: the model was initially constructed with five compartments (including POM and MAOM substrates and microbial biomass, as well as DOM), but assuming that microbial biomass and DOM are at quasi-equilibrium allows reducing the model to two compartments. This simplification has minor consequences on the POM and MAOM dynamics as long as both microbial biomass and DOM turn over faster than the POM and MAOM substrates. Microbial biomass has a turnover time in the order of a few months (Spohn et al., 2016) and DOM dynamics are even

faster—shorter than the turnover of POM and MAOM. Therefore, our quasi-equilibrium assumption appears to be reasonable.

-    Organic matter chemical heterogeneity: residues + POM and MAOM contain compounds with contrasting chemical characteristics (depending on residue chemistry and on the pathway of stabilization into MAOM, respectively), but we neglected these chemical differences both to keep the model simple and because of limited data to parameterize

more than one compartment for residues + POM and one for MAOM. As a consequence, we also neglected the decreasing rates of decomposition through time as a result of accumulating recalcitrant compounds. However, we can expect that less decomposable compounds remain in both POM and MAOM (because of their different chemical recalcitrance or accessibility, respectively), so that the ratio of the decay constants for these compartments (i.e., parameter $\kappa$) should remain relatively stable, which is the only assumption we need to make in our derivation.

Therefore, neglecting chemical heterogeneity may significantly affect the prediction of decomposition rates, but it is likely to be less important when modelling residues + POM and MAOM in phase space.

-    Microbial traits: microorganisms growing on POM are likely different from those feeding on organic matter desorbed from minerals. For example, we could expect higher fungal to bacterial ratio in residues + POM, with higher microbial biomass C:N and possibly lower CUE (Soares and Rousk, 2019), but lack of specific information on microbial traits

within the soil fractions does not allow us to parameterize these communities in the model (though soil fraction-specific traits are retained in the general solutions of the mass balance equations).

## 4.3. Reconciling contrasting decomposition patterns in phase space

Our phase space representation of residues + POM and MAOM dynamics highlights a simple and consistent pattern—as residues are decomposed, the residues + POM compartment is depleted, while MAOM gains C and N (early phase of

decomposition). However, also residue-derived C and N in MAOM will be decomposed eventually (late phase). These

processes lead to a humped relation between the fractions of residue C or N recovered in MAOM and the C fraction recovered in POM (Fig. 3). The shape of this relation depends on microbial CUE and the partitioning of C and N between in vivo and ex vivo pathways (Fig. 2). Generally, higher CUE and ex vivo stabilization promote C and N accumulation in MAOM (steeper increase in MAOM as residues + POM is reduced). In both cases, this is due to lower C losses via respiration in the residues + POM compartment promoting C (and N) transfer to MAOM and retention in stabilized form in that compartment.

Because of the infrequent sampling in the incubation studies, the whole pattern of increasing and decreasing MAOM has not been observed so far. For example, C in MAOM increased through time in some studies (Cheng et al., 2023; Fulton-Smith and Cotrufo, 2019; Leichty et al., 2021; Neupane et al., 2023), but in others it decreased (Su et al., 2020; Throckmorton et al., 2015; Wang et al., 2017). N in MAOM tends to increase through time in most studies (Fulton-Smith and Cotrufo, 2019; Mitchell et al., 2018; Nunez et al., 2022), but it can also decrease (Kölbl et al., 2006). Our model links through analytical equations these two regimes of early decomposition associated to transfer to MAOM and late decomposition associated to destabilization from MAOM. These equations allow to compare datasets that might appear inconsistent at first sight.

The phase space representation also shows that the stoichiometry of residues + POM is conserved during residue decomposition and stabilization, regardless of the residue initial C:N ratio (Figure 3A). This result might seem surprising, as N is preferentially retained during residue decomposition, often resulting in a temporary net N accumulation if residues are N poor (Moore et al., 2006; Parton et al., 2007). This pattern can be explained by microbial N immobilization and recycling of N rich microbial necromass within the residues, which gradually lowers the residue C:N to values close to those of microbial biomass (Manzoni et al., 2008). The observation that the residues + POM compartment retains the initial residue C:N ratio indicates that microbial necromass is not recycled within that compartment, but rather it is stabilized into the MAOM fraction through the in vivo pathway (parameter $m = 1$). Therefore, the phase space of C and N in residues + POM both informs about stabilization mechanisms and helps constrain model parameters.

### 4.4. What is the dominant pathway of C and N stabilization in MAOM?

Earlier studies identified the origin of MAOM using microbial biomarkers (e.g., amino sugars) that trace microbial necromass contributions to MAOM, molecular fingerprinting to partition MAOM into microbial- or plant-derived based on their specific molecular signatures, or isotopic and stoichiometric mixing models (Chang et al., 2024; Whalen et al., 2022). Leveraging the contrast in N contents of microbial biomass (N-rich) and plant residues (N-poor), Chang et al. (2024) estimated that between 34% and 47% of MAOM is of microbial origin. Estimates based on amino sugar analysis can be similar or lower (Whalen et al., 2022). Our estimates suggest that approximately 75% of MAOM C and almost all MAOM N are formed thanks to the in vivo pathway. It is possible that the contribution of the in vivo pathway we estimated is higher because we did not consider the stabilization of residues + POM within very fine aggregates (Mueller et al., 2012), which would be separated as MAOM. Another explanation may be that we neglected the stabilization of dissolved C and N at the very beginning of decomposition. Our model can account for this process, but our data analysis to test its relevance was not conclusive (Supplementary Information, Section S3). A third plausible explanation is that the persistence of necromass and other sources of MAOM differ,

so that despite a larger contribution of the in vivo pathway (predicted by our model), compounds stabilized via the ex vivo pathway could persist longer in the MAOM compartment. This would result in lower percentages of microbe-derived MAOM as estimated by Chang et al. (2024). This explanation appears plausible in the light of relatively short turnover time of necromass in MAOM (< 1 year, Buckeridge et al., 2022) compared to the bulk MAOM. Therefore, we conclude that the stabilization of residue C and N in MAOM is dominated by the in vivo pathway, but we also acknowledge that other sources of C and N that would contribute ex vivo were not considered in the isotope tracing experiments or in our model.

### 4.5. What are the drivers of the stabilization pathway?

Our results show that a higher clay fraction is associated with more dominant in vivo stabilization of both C and N ($f_C$ and $f_N$ in Fig. 6). This is consistent with empirical evidence that the in vivo pathway is promoted in finer textured soils (Chang et al., 2024), and thus supports the idea that in these soils, depolymerization products are used by microorganisms whose necromass is eventually stabilized. Finer textured soils can promote microbial growth and necromass production by improving moisture retention besides offering more available minerals for stabilization of the microbial products (Mao et al., 2024). Similar to Chang et al. (2024), we found negative effects of SOC on both $f_C$ and $f_N$, indicating that in organic matter rich soils the in vivo stabilization pathway is less important than in organic matter poor soils. The result that stabilization through the in vivo pathway is more important in clay rich soil, but less so in C rich soils suggests that in vivo stabilization is particularly sensitive to saturation of mineral surfaces (Georgiou et al., 2022). This finding is consistent with N-rich organic matter—likely of microbial origin—directly bonding to minerals (Spohn, 2024) and thus being dependent on the availability of active mineral surfaces. In contrast, C-rich organic matter from the ex-vivo pathway tends to indirectly bond to minerals through organic matter-organic matter interactions (Spohn, 2024) and is thus less constrained by saturation of the mineral surfaces (Begill et al., 2023).

According to the Microbial Efficiency-Matrix Stabilization (MEMS) hypothesis, labile and N rich residues would be more likely to be stabilized via the in vivo pathway, thanks to more efficient conversion of residue-derived C and N into biomass (Cotrufo et al., 2013). The general trend of decreasing CUE as residue C:N increases (Manzoni et al., 2008, 2017) was confirmed here (Fig. 6a), and low residue C:N indeed promoted stabilization via the in vivo pathway, but only in soils with more than about 15% clay content (Fig. 6c).

The in vivo pathway was also promoted by warmer conditions (although the temperature effect was only marginally significant)—again consistent with the results by Chang et al. (2024).

## 5. Conclusions

We proposed a simple diagnostic model to interpret data on residue incorporation into POM and MAOM. The model is solved analytically in the phase space—i.e., by expressing one variable as a function of other variables instead of time. This approach
moves away from the usual focus on kinetics and allows quantifying the partitioning of C and N between two main pathways of stabilization: in vivo stabilization of microbial necromass and ex vivo stabilization of depolymerization products. We found that the majority of C and N derived from added residues is stabilized through the in vivo pathway. This pathway is particularly dominant in clay rich and C poor soils, where stabilization is less limited by saturation of the mineral surfaces. Overall, these findings support the idea that a large fraction of MAOM is derived from microbial necromass, but also that the availability of
mineral surfaces affects the relevance of this stabilization pathway.

## Appendix A

General trends in MAOM accumulation can be assessed by calculating the change in C or N in MAOM per unit change in residues + POM ($\Delta c_M \Delta c_P^{-1}$). Because the residues + POM compartment loses mass due to decomposition, its changes are always negative ($\Delta c_P < 0$). As a consequence, negative relative changes in MAOM ($\Delta c_M \Delta c_P^{-1} < 0$) indicate accumulation of mass in MAOM. Generally, in the early phase of decomposition when the fraction of remaining residues + POM is still high, both C and N accumulated in MAOM, but below a $c_P$ or $n_P$ threshold, both C and N are lost from MAOM (Fig. A1). The turning points when MAOM starts being depleted are at $c_P \approx 0.18$ and $n_P \approx 0.42$ (i.e., earlier than for C). We note that fewer data on N accumulation in MAOM do not allow to constrain this threshold as accurately as for C accumulation.

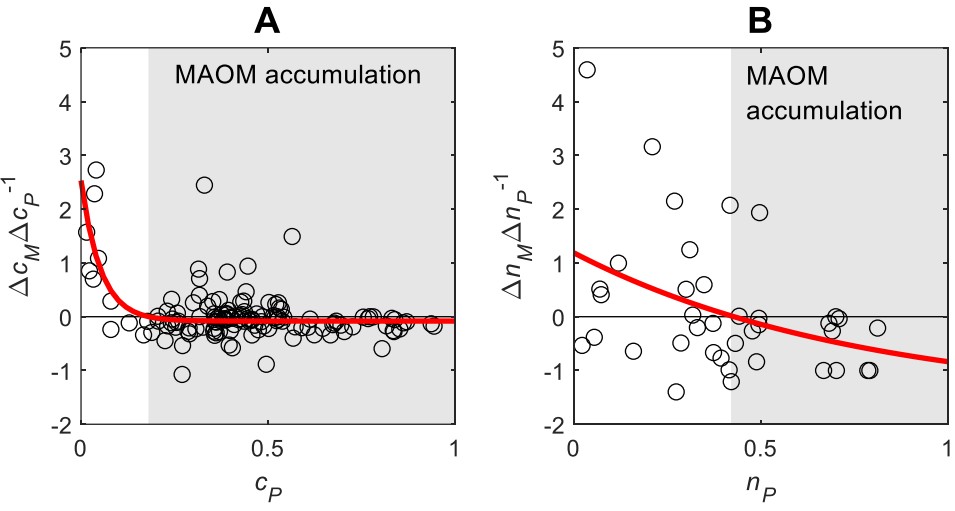

**Figure A1. Changes in C or N in MAOM per unit change in C or N in residues + POM, as a function of remaining C or N in residues + POM (panel A for C and panel B for N). $c_P$ and $c_M$ ($n_P$ and $n_M$) denote the fractions of remaining C (respectively N) in residues + POM and MAOM. Time progresses from right to left as $c_P$ and $n_P$ decrease. Data points are from all datasets containing at least two subsequent measurements for the same treatment, site, and residue type. Solid curves are fitted exponential functions with an asymptote, used to define the threshold $c_P$ and $n_P$ at which MAOM accumulation (light gray) turns into depletion (we excluded from the regression outliers defined as values lower than the 3th and higher than the 97th percentile).**

## Data availability

Data on residue-derived carbon and nitrogen in the residue, dissolved, particulate, and mineral associated organic matter compartments are deposited in the open access Bolin Centre Database (Manzoni et al., 2024).

## Author contribution

SM collated and analyzed data, developed and implemented the model, and wrote the first draft; FC contributed to data interpretation and model development, and commented and edited the manuscript.

## Competing interests

The authors declare that they have no conflict of interest.

## Funding

This project has received funding from the European Research Council (ERC) under the European Union's Horizon 2020 Research and Innovation Programme (grant agreement no 101001608). This cooperation was facilitated by the August T. Larsson Foundation who supported M.F. Cotrufo's guest professorship at the Swedish University of Agricultural Sciences (Uppsala, Sweden).

## Acknowledgements

We thank for providing raw or unpublished data and for help interpreting published data: Kate Buckeridge (Luxembourg Institute of Science and Technology), Yan Duan (Chinese Academy of Sciences, Nanjing), Gabriel Dias Ferreira (Colorado State University), Ed Gregorich (Carleton University), Michelle Haddix (Colorado State University), Jian Jin (La Trobe University), Xinchang Kou (Chinese Academy of Sciences, Shenyang), Joanna Ridgeway (West Virginia University, Morgantown), Alin Shen (Zhejiang Academy of Agricultural Sciences), Xiaoke Zhang (Chinese Academy of Sciences, Shenyang).

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
