# Peer review of "Mechanisms of soil organic carbon and nitrogen stabilization in mineral associated organic matter – Insights from modelling in phase space"

_EGUsphere, 2024_

## Author Comment (AC2)

**Responses to Reviewer #1**

We thank Reviewer #1 for their comments. Our responses are reported below in normal font, while the reviewer's comments are in *italic*.

*This study proposes a phase-space model fitted to about 40 studies which estimates that ~75% of MAOM C and nearly all of MAOM N are stabilized by the in vivo pathway, in contrast to measurements which tend to be lower than this. The authors posit a few reasons for the discrepancy, including ex vivo stabilization pathways missing from the model, and the most interesting which is that necromass may be stabilized faster but turns over at a faster rate, resulting in less persistent MAOM from necromass.*

We agree that some results might be surprising, but can still be explained in light of current conceptual understanding. Besides the different fraction of carbon (or nitrogen) stabilized through the in vivo pathway, in our view it is particularly interesting to see trends with soil texture and organic matter content that are consistent with the idea of saturation of mineral surfaces. Overall the fractions we estimated might be different due to methodological reasons, but the underlying trends are quite insightful.

*I think in order to explore the hypotheses brought up by this study it is useful to run the model considering time. I assume that you would get the same curves that can be seen in the main results (Figs 2 and 4) when running the model forward as differential equations and plotting the model output as the relationship between pools, so I'm not sure what the phase space simplification really adds.*

Solving the mass balance equations through time and then plotting one state variable as a function of the other will give the same results as directly solving the system in the phase space (see an example in the response to Reviewer #2). In practice the two approaches lead to the same modelled trajectories, and if the interest is in considering time dynamics, then the first approach should be used. However, the phase space approach has complementary advantages, such as: i) in the phase space, differences in the reaction kinetics do not interfere with the estimation of partitioning coefficients (in our model $l$ and $m$) that provide information on the stabilization pathways, and ii) the analytical solutions in phase space are sufficiently compact to allow for mathematical insights that would not be possible when solving the model through time. Moreover, a fully analytical solution through time might not be possible if the decomposition kinetics are nonlinear, requiring numerical solutions. Here we do not need to make any assumptions on the decomposition kinetics (besides a proportionality argument), leading to very general analytical solutions in phase space. We discuss these advantages in Section 4.1. We will also address this reviewer's comment by revising Figure 4 (as reported below in Figure R1) and elaborating further on this point in the text in reference to the figure.

*I think that retaining the ability to consider absolute values of MAOM and POM and track losses (C and N min) are relevant for addition scenarios in the field where total C retained in the system is as or more relevant than the form of C.*

Normalizing the C and N contents in the modelled pools by the added residue C and N does not preclude the possibility to calculate absolute amounts of C and N in the soil pools. The analytical solutions can be simply multiplied by the added mass of C and N to obtain the absolute amounts, which would be expressed as contents, concentrations, or stocks depending on the units used for the added residue mass. We can explain by adding just before Eq. (10) where the normalization step is presented: "Moreover, the equations expressed in normalized form are independent of the units used to quantify inputs and mass in each compartment, and if needed it is easy to convert the normalized variables into absolute quantities by multiplying by the mass of added residue C."

*There were similarly some places in the text (L357-359) and Fig. 4 where I was really curious how accumulation of N vs C in MAOM was evolving over time.*

We are also curious about C and N accumulation in the MAOM compartment, but the data do not allow a consistent assessment across studies. Each study was designed differently, so sampling started and ended at different times (and different degrees of residue decomposition) and sampling frequency also varied (between 1 and 6 sampling times). Due to the low temporal resolution of the measurements, in most datasets the peak in residue-derived C in MAOM (before being decomposed) is not apparent from the data. Moreover, environmental conditions during the incubations (in the field or in the lab) also differed. Unfortunately, these differences among datasets and in general data limitations do not allow a systematic time series analysis to identify temporal trends in MAOM accumulation.

However, in a revised manuscript we can expand Figure 4 as shown below (Figure R1) to also illustrate the time series of the data, as suggested by the reviewer. As shown in panels A-F of Figure R1, the temporal trends in the data are highly variable (on purpose in this revised figure we would leave the same axis scaling in each row to highlight differences among studies), compared to the clearer patterns emerging in phase space (panels G-I). Other examples are shown below in response to another comment (Figure R3). We could use the expanded Figure 4 to illustrate the point that phase space representations allow to synthesize data more effectively than in the usual time domain, also addressing the reviewer's general comment above.

[Figure]

**Figure R1. Examples of data time series from incubations (A-F) and model fitting in the phase space (G-I), for residues with increasing residue C:N from the left to the right column: A-C) fractions of added C in residues + POM ($c_P$, green circles) and MAOM ($c_M$, brown circles) as a function of time; D-F) fractions of added N in POM + residues ($n_P$, green triangles) and MAOM ($n_M$, brown triangles) as a function of time; G-I) $c_M$ (black circles) and $n_M$ (gray triangles) as a function of $c_P$. In G-I, we fitted parameters $e$ and $l$ in the functions $c_M(c_P)$ and $n_M(c_P)$ with $m = 1$ and $\kappa = 0.05$ (Table 2). Data are from: A, D, G) Buckeridge et al. (2022) (residues:** *Escherichia coli* **necromass, C:N=3.4), B, E, H) Mitchell et al. (2018) (residues:** *Chloris gayana*, **C:N=14.2), and C, F, I) Lavallee et al. (2018) (residues:** *Andropogon gerardii*, **silt soil, C:N=28.2). In G-I, residue decomposition progresses from right to left along the curves, as $c_P$ decreases; the dot-dashed 1:1 lines represent equality between the fractions of added C or N shown on the y-axes and $c_P$ shown on the x-axes.**

More in general, in a revised manuscript we can assess to what degree C and N in MAOM increase or decrease in relation to the fraction of remaining C in residues + POM. This analysis would show accumulation pathways as the reviewer suggests, but without (or with less) variability induced by environmental conditions across studies. In practice, such trends can be evaluated numerically from pairs of data points acquired in subsequent times, by computing the change in MAOM (C or N) per unit change in residues + POM (C or N). These local slopes of the MAOM vs. residues + POM relations can then be plotted as a function of residues + POM to identify MAOM accumulation and depletion phases. A negative slope indicates accumulation of MAOM as residues + POM are depleted, whereas a positive slope indicates a depletion of both MAOM and residues + POM. Overall, the slopes are negative at high values of the fraction of residues + POM (early decomposition phase), but they turn negative at low values of the fraction of residues + POM (late decomposition phase) (Figure R2). These trends are consistent for C (Figure R2A) and N (Figure R2B). The turning points when MAOM starts being depleted are at a fraction of remaining C in residues + POM of ≈0.26 and at a fraction of remaining N in residues + POM of ≈0.18. This new figure would be included in the Appendix and referred to in Section 3.3, where examples of time series will be shown. We hope that such an analysis can at least partly satisfy readers interested in residue-derived MAOM accumulation and depletion.

[Figure]

**Figure R2. Changes in MAOM per unit change in residues + POM, as a function of remaining residues + POM, for A) residue C and B) residue N. As in the submitted manuscript, $c_P$ and $c_M$ ($n_P$ and $n_M$) denote the fractions of remaining C (respectively N) in residues + POM and MAOM. Time progresses from right to left as $c_P$ and $n_P$ decrease. Data points are from all datasets containing at least two subsequent measurements for the same treatment, site, and residue type. Solid curves are fitting exponential functions with an asymptote (we excluded from the regression outliers defined as values lower than the 3th and higher than the 97th percentile). The MAOM accumulation phase is highlighted in light gray.**

*I suppose my ask to the authors is to justify to readers the utility of phase space modeling compared to running the time-dependent form of the model, or to use time-dependent modeling to more completely address some of the hypotheses advanced, especially about N vs C accumulation in MAOM.*

Thanks for this suggestion, we are happy to provide this justification as also explained above. However, we believe that analyzing soil fraction data in phase space helps focusing on intrinsic properties. To illustrate them for the interest of the reviewer we provide a couple of examples (Figure R3). In one example, chemically different litter types are incubated in the same conditions (*Sorghum* residues), whereas in the second example the same litter type is incubated with or without additional mineral N (*Miscanthus* residues). In both cases, the temporal dynamics of residues + POM (Figure R3A) and MAOM (Figure R3B) exhibit large variability due to either the different litter chemistry or the incubation conditions, but the trajectories in the phase space converge to similar patterns (see orange and blue lines or yellow and green lines in Figure R3C). We would not include Figure R3 in a revised manuscript, as the same point is already illustrated in the proposed Figures R1 and R2.

In synthesis, we agree with the reviewer that the temporal dynamics contain additional information, but that will mostly reflect incubation conditions rather than the intrinsic capacity of the soil and its microbial community to promote organic matter stabilization in MAOM through the in vivo or ex vivo pathway. A study focusing on the temporal dynamics would be useful to test the effect of soil moisture or temperature on the kinetics of the stabilization, or to compare field-based to lab-based incubations, but these questions are outside our scope in this contribution. The two more conceptual questions we stated in the Introduction (numbered ii and iii) focus on C and N partitioning between pathways rather than on the kinetics of the process: ii) What is the dominant pathway of C and N stabilization in MAOM? iii) What are the drivers of the stabilization pathway as represented by model parameters?

Therefore, we will be happy to revise the manuscript as suggested above to better motivate our choice to focus on the phase space rather than the time domain, but we would be reluctant to fit the model to the time series as interpreting patterns in both kinetic constants (which we now do not consider) and partitioning coefficients (our current focus) would be difficult and results would depend on the selected kinetics.

[Figure]

**Figure R3. Examples of time trajectories of the fractions of remaining C in either A) residues + POM or B) MAOM; C) phase space representation of the same data—i.e., fraction of remaining C in MAOM as a function of the fraction of remaining C in residues + POM. Colors and symbols refer to different datasets: circles refer to shoots (blue) and roots (orange) of *Sorghum bicolor* (Fulton-Smith and Cotrufo, 2019); squares refer to roots of *Miscanthus sacchariflorus* incubated with additional mineral N (yellow) or without mineral N (green) (Poeplau et al., 2023).**

*L59-60: I think this sentence needs a main clause verb.*

Thanks for catching this mistake. Deleting "and" fixes the problem.

*L120: Probably good to acknowledge that some C will be lost as DOC leaching, even if it is small and you don't consider it here.*

We can clarify by adding that "Leaching of dissolved organic matter is neglected."

*Fig 1: A fraction of necromass is sorbed and the rest goes back into particulate C... what is that fraction based on?*

The fraction of necromass that is sorbed is quantified by the parameter $m$, which is estimated by fitting the model to the data. Therefore, we do not make any assumptions about this fraction, but rather let the data tell if it is large or small. We would not add a clarification on this comment, because the estimation approach for parameter $m$ is presented in Section 2.2.2.

*At L322, Section 0 is referenced as where you explain how m is constrained but I could not find this section.*

Sorry about this, there was an issue with a section referencing. It should read "Section 3.2"

*Eq.10 is the same as Eq. 9 but with a different boundary condition. I would suggest referencing Eq. 9 instead of printing the equation again.*

Equation (10) looks similar, but besides the different boundary conditions, it also includes normalized variables (small $c$ instead of capital $C$), so we would prefer to keep it.

*N mineralization assumes a one-way flow where N is mineralized but not taken up again. This is similar to a lot of soil models but not necessarily realistic, so perhaps worth acknowledging that in text somewhere.*

We understand the concern, but the reviewer missed that this is not necessarily the case in our formulation of net N mineralization, which accounts for possible net N immobilization when microbial N demand is higher than the N supply from organic matter decomposition. We can add a clarification below Eq. (17): "These formulations for net N mineralization allow capturing both net N release if substrates are sufficiently rich in N ($N_{PS}/C_{PS} > e_P r_{PB}$, $N_{MS}/C_{MS} > e_M r_{MB}$) and net N immobilization when they cannot provide enough N for microorganisms ($N_{PS}/C_{PS} < e_P r_{PB}$, $N_{MS}/C_{MS} < e_M r_{MB}$)."

*Table S1: If POM+residues are considered as one category in the model (Fig 1) then why are they separated out here? Please clarify how each category is allocated when used in the model.*

The dataset contains data on residues alone, POM alone, or combined residues + POM. As most datasets report combined residues + POM, we chose to use this as a state variable in the model. We can clarify in Section 2.1.1 "The choice of merging these two fractions in one model compartment is motivated by the fact that in many datasets residues and POM were not separated." Further details on the type of data used and data processing are given in Section 2.2.1.

*L266 seems to imply that those studies without POM+residues reported were not used, does this change the reported sample size used in analyses?*

Six studies reporting only POM or only residue C were not used. The reviewer is correct that the total number of studies mentioned in the Abstract and Methods should reflect only the number of studies actually used. We can update those values after we include the additional dataset suggested below.

*L357-359: I found it curious that a higher C:N ratio causes lower C accumulation in MAOM but higher N accumulation. Does the C:N of MAOM really decrease in this scenario or is this an artifact of only viewing Cm as relative to Cp which is also changing over time?*

Higher residue C:N does not affect C accumulation in MAOM (dashed and solid curves overlap in Figure 4, top left panel); however, higher C:N does promote N accumulation in MAOM because more N is immobilized by microorganisms, converted into necromass, and eventually transferred from the residues + POM to the MAOM compartment.

*In Figure 4, it is also interesting that N seems to accumulate faster than C in MAOM, but again we are not looking at time here so travel along either curve could occur at different speeds. Can you discuss this or provide some information about the time component?*

The reviewer is correct—we are not looking at temporal trajectories, so it is not possible to read reaction speed in these phase spaces. As suggested above, we can add panels in Figure 4 to illustrate the temporal changes (Figure R1) and comment on those patterns, for example by highlighting that "more N than C accumulate in MAOM at a given time or $c_P$ value (compare Figure 4B and 4E or Figure 4C and 4F), indicating preferential retention and stabilization of N when residues with high C:N are decomposed (as also shown in Figure 2)."

*Following from the understanding that in vivo is more important for N than C, I wonder if this can tell us something about the rate of the in vivo pathway vs ex vivo.*

We appreciate the reviewer curiosity, but as extensively explained above with the current analysis, we cannot evaluate and rank the rates of C and N stabilization, but only the relative proportions.

*Figure 3b: I know there are many data sources, but can the data source color bar be discretized?*

Yes, we can replace the existing Figure 3B with Figure R4. However, the difference with respect to the original figure is minimal given the number of discrete levels.

[Figure]

**Figure R4. Revised panel B of Figure 4 in the original manuscript, with discretized colorbar.**

*L466: I was a little surprised to see such a strong prediction of clay trends in B and C but no trend in univariate space (Fig 5). Do you feel confident in the LME prediction?*

P-values for the clay effects on the fraction of C and N following the in vivo pathway are low ($<0.05$) and coefficients are relatively large, so yes, we are confident there is a trend. If invited to revise the manuscript, we can also strengthen our analysis by adding some datasets we found during the review phase (Hilscher and Knicker, 2011; Córdova et al., 2018). To address this comment and similar one by Reviewer #3, we propose to remove the univariate analysis—it is probably more confusing than helpful.

**References**

Buckeridge, K., Mason, K., Ostle, N., McNamara, N., Grant, H., Whitaker, J., 2022. Microbial necromass carbon and nitrogen persistence are decoupled in agricultural grassland soils. Communications Earth & Environment 3. doi:10.1038/s43247-022-00439-0

Córdova, S., Olk, D., Dietzel, R., Mueller, K., Archontouilis, S., Castellano, M., 2018. Plant litter quality affects the accumulation rate, composition, and stability of mineral-associated soil organic matter. Soil Biology & Biochemistry 125, 115–124. doi:10.1016/j.soilbio.2018.07.010

Fulton-Smith, S., Cotrufo, M., 2019. Pathways of soil organic matter formation from above and belowground inputs in a Sorghum bicolor bioenergy crop. Global Change Biology Bioenergy 11, 971–987. doi:10.1111/gcbb.12598

Hilscher, A., Knicker, H., 2011. Degradation of grass-derived pyrogenic organic material, transport of the residues within a soil column and distribution in soil organic matter fractions during a 28 month microcosm experiment. Organic Geochemistry 42, 42–54. doi:10.1016/j.orggeochem.2010.10.005

Lavallee, J., Conant, R., Paul, E., Cotrufo, M., 2018. Incorporation of shoot versus root-derived 13C and 15N into mineral-associated organic matter fractions: results of a soil slurry incubation with dual-labelled plant material. Biogeochemistry 137, 379–393. doi:10.1007/s10533-018-0428-z

Mitchell, E., Scheer, C., Rowlings, D., Conant, R., Cotrufo, M., Grace, P., 2018. Amount and incorporation of plant residue inputs modify residue stabilisation dynamics in soil organic matter fractions. Agriculture Ecosystems & Environment 256, 82–91. doi:10.1016/j.agee.2017.12.006

Poeplau, C., Begill, N., Liang, Z., Schiedung, M., 2023. Root litter quality drives the dynamic of native mineral-associated organic carbon in a temperate agricultural soil. Plant and Soil. doi:10.1007/s11104-023-06127-y

---

## Author Comment (AC3)

**Responses to Reviewer #2**

We thank Reviewer #2 for their comments. Our responses are reported below in normal font, while the reviewer's comments are in *italic*.

*Manzoni and Cotrufo propose a two-compartment model (POM + residues and MAOM), which can be used to diagnose through which pathway (in vivo vs. ex vivo) C and N are stabilized. The simplicity of the model reduces the number of parameters and allows the authors to solve their equations analytically in the so-called space phase, i.e. they focus on relations between compartments instead of analyzing temporal trajectories to detect underlaying mechanisms. The model suggests that stabilization via the in vivo pathway is more relevant in clay-rich soils and less relevant in C-rich soils, whereby the calculated fractions seem to be higher than reported by earlier studies. Overall, the paper is well (and lively!) written and the chosen approach is an interesting method to explain potentially appearing inconsistencies in data sets. E.g. the model is able to capture MAOM stabilization during early decomposition and destabilization of MAOM in late decomposition by the same mechanism. From that perspective and because it adds a new insight into the process of C and N stabilization, the paper is a highly relevant for the BG community. Also, discrepancies with earlier studies are mentioned and discussed appropriately.*

We thank the reviewer for their supportive comments—both on our writing style and on the content of the manuscript.

*However, I'm concerned about the assumptions and approximations, which are made during the derivation of the solution, and that are a bit loosely justified (see below), and would ask the authors to elaborate these parts a bit further.*

We agree that assumptions need to be spelled out clearly—please see our responses below.

*As mentioned before, I'd ask the authors to explain used assumptions and approximations in more detail. Especially the assumption of (quasi-)equilibrium needs more explanation and discussion. Why (under which circumstances) can quasi-equilibrium be assumed?*

Reviewer #3 had a similar comment regarding our assumption that microbes at quasi-equilibrium, so we respond to both comments in the same way.

To assess the consequences of our quasi-equilibrium assumption, we modelled (numerically) both substrates and microbial biomass in residues + POM and in MAOM (Eq. 1-2 and 5-6 in the main text), as shown in Figures R1 and R2 (denoted as 'full model'). For these model runs, we selected parameters reflecting the findings in the submitted manuscript and thus representative of the soil incubations we analyzed, but—different from the analytical model presented in the manuscript—we also had to specify kinetic parameters for residues + POM and microbial biomass. For the former we assumed residues + POM with a 6 month turnover time (reasonable for labile residues), whereas for the latter we considered two turnover rates (6 or 2 months, consistent with the range estimated for surface soils by Spohn et al., 2016). With the lower microbial mortality, there are some discrepancies between the full model and the minimal model presented in the manuscript—specifically, C accumulates in microbial biomass before being released and stabilized in MAOM in the full model, different from the fast transfer of C to MAOM in the minimal model (Figure R1B). This correspond to lower curves in the phase space—i.e., lower C in MAOM for a given C in residues + POM (Figure R1C). Thus, in the presence of microbial biomass with relatively slow turnover, we might be overestimating MAOM accumulation. However, this error decreases with faster microbial turnover (Figure R2A, B), or with slower decomposition rate of residues + POM (not shown). We can thus conclude that our approximation that microbes are in quasi-equilibrium is reasonable except when both microbial turnover is slow and residue turnover is fast.

Adding these figures in a revised manuscript does not seem necessary, but we would include this discussion on the quasi-equilibrium approximation in a new section of the Discussion "4.2. Model limitations." In this new section, we would explain: "The model was constructed with five compartments (including POM and MAOM substrates and microbial biomass, as well as DOM), but assuming that microbial biomass and DOM are at quasi-equilibrium allows reducing the model to two compartments. This simplification has minor consequences on the POM and MAOM dynamics as long as both microbial biomass and DOM turn over faster than the POM and MAOM substrates. Microbial biomass has a turnover time in the order of a few months (Spohn et al., 2016) and DOM dynamics are even faster—shorter than the turnover of POM and MAOM. Therefore, our quasi-equilibrium assumption appears to be reasonable."

[Figure]

**Figure R1. Temporal trends in the fractions of added C in: A) residues + POM (microbial biomass, $c_{PB}$, and total, $c_P$) and B) MAOM (microbial biomass, $c_{MB}$, and total, $c_M$). C) Phase space representation of the same time trajectories—i.e., $c_M$ is plotted as a function of $c_P$.** In all panels, curves refer to the 'full model' where microbial biomass dynamics are included (solid lines), and the 'minimal model' where microbial biomass is assumed in quasi-equilibrium (dashed lines). In panel C, the minimal model is solved both numerically (black dashed line) and analytically as in the submitted manuscript (gray thick dashed line). Parameters: $b$=1, $m$=1, $l$=0.1, $e$=0.2, $\kappa$=0.05; we also assumed first order kinetics for residues + POM decomposition, with decay constant of 2 y$^{-1}$, and for microbial mortality, with rate constant of 2 y$^{-1}$.

[Figure]

**Figure R2. Same as Figure R1, but with a microbial mortality rate constant of 6 y⁻¹.**

*Are the data filtered by that?*

No, there is no way from the data available to filter datasets according to the turnover rate of microbial biomass or DOC. Our arguments for model simplification are general and not data-driven.

*How does this relate to the later discussion about "early" and "late" decomposition?*

Early and late decomposition phases are discussed based on residue + POM decomposition and MAOM accumulation and subsequent decomposition, which are both evident from the data and model results even when assuming that microorganisms are in quasi-equilibrium.

*How do this assumption and the other assumptions and approximations, which are used to derive the final analytically-solvable model, i.e. the first-order decay with a constant decomposition rate or the "similar" C:N ratios, affect model results and limit their transferability, e.g. under a changing climate?*

We would like to first clarify that we did not assume first-order decay, but only that the decay rate constants for residue + POM and MAOM decomposition are proportional. This assumption is less strict than assuming first-order decay for these processes. Regarding our assumption of a single value for microbial biomass C:N ratio, we could not find evidence in the literature of different microbial C:N between POM and MAOM fractions. It is possible that microbial communities associated with POM are more fungal dominated and thus have higher C:N, but we can only speculate about the extent of this difference. We can elaborate on the various model assumptions—including these ones—in a new Discussion section "4.2. Model limitations."

In this new section we would explain: "microorganisms growing on POM are likely different from those feeding on organic matter desorbed from minerals. For example, we could expect higher fungal to bacterial ratio in POM, with higher microbial biomass C:N and possibly lower CUE (Soares and Rousk, 2019), but lack of specific information on microbial traits within the soil fractions does not allow us to parameterize these communities in the model (though soil fraction-specific traits are retained in the general solutions of the mass balance equations)."

*Figure 2: I think, it would help the reader, if you could add a reference figure (base values used) to understand the overall model behavior and the direction of the changes. I would also like to see the varied values, i.e. the values that are used to derive the solid and dashed lines in each sub-figure, in the figure description.*

We can revise Figure 2 according to this suggestion, by adding a column with model results obtained using the baseline parameter values (Figure R2). The values of the parameters used to make each panel are listed in the legends at the bottom of the figure, so it seems redundant to also report them in the caption.

[Figure]

**Figure 2. Fraction of added C in MAOM, $c_M$ (top row), fraction of added N in residues + POM, $n_P$ (center row), and fraction of added N in MAOM, $n_M$ (bottom row), as a function of the fraction of added C in residues + POM, $c_P$, under different stabilization pathway scenarios (colors) and when varying the values of model parameters around baseline values shown in the 1st column: residue N:C ratio, $r_0$ (2nd column), microbial carbon use efficiency, $e$ (3rd column), and ratio between the decay constants of MAOM and residues + POM decomposition, $\kappa$ (4th column). Three stabilization scenarios are considered: dominant ex vivo stabilization ($l = 1$; orange), dominant in vivo stabilization ($l = 0, m = 1$; blue), and a combination of pathways denoted by 'mix' ($l = 1/2, m = 1/2$; green). In all panels, residue decomposition progresses from right to left along the curves, as $c_P$ decreases. The dot-dashed black lines indicate 1:1 lines, which represent equality between the fractions of added C or N shown on the y-axes and $c_P$ shown on the x-axes; the added residues are assumed to be insoluble ($b = 1$).**

*What is "Section 0"? (L322, L416)*

These are issues with section referencing. They should read "Section 3.2" and "Section 3.4", respectively.

*L34 Please check the unit.*

Font issue—it should be µm.

*Table 2: I personally would like to have the Parameter/Symbol entries all centered (or maybe all right-bounded), but not changing within lines, which seems to happen for all entries that do not have subscripts. Or is there I reason for that?*

No particular reason, but it seems equation objects sit in the center by default despite normal text being aligned to the left. The table can be easily formatted so that all symbols are centered.

**References**

Soares, M., Rousk, J., 2019. Microbial growth and carbon use efficiency in soil: Links to fungal-bacterial dominance, SOC-quality and stoichiometry. Soil Biology and Biochemistry 131, 195–205. doi:10.1016/j.soilbio.2019.01.010

Spohn, M., Pötsch, E.M., Eichorst, S.A., Woebken, D., Wanek, W., Richter, A., 2016. Soil microbial carbon use efficiency and biomass turnover in a long-term fertilization experiment in a temperate grassland. Soil Biology and Biochemistry 97, 168–175. doi:10.1016/j.soilbio.2016.03.008

---

## Author Comment (AC4)

**Responses to Reviewer #3**

We thank Reviewer #3 for their comments. Our responses are reported below in normal font, while the reviewer's comments are in *italic*.

*The submitted manuscript aims at improving our understanding of the formation of mineral associated organic matter (MAOM), particularly the relative contributions of plant versus microbial derived carbon to MAOM formation. Therefore, they apply the innovative approach of fitting respective parameters of a simple soil microbial model in phase space. The datasets used are from decomposition experiments, where added labelled residues are traced from the particulate organic matter (POM) to the MAOM fraction. Instead of looking at temporal changes, the relative proportional changes of POM and MAOM along the decomposition continuum are mathematically formulated and solved, considering both C- and N- dynamics. The results showed that the in-vivo pathway (microbial derived) was more important than the ex-vivo pathway (plant-derived) MAOM-C and particularly MAOM-N formation. It was further possible to identify some controls on the relative importance of the two pathways. The authors conclude that the in vivo-pathway is particularly important in clay rich and carbon poor soils and that this path is accordingly stronger controlled by available mineral surfaces. The authors did a great job in writing and explaining each modelling step in the manuscript and I enjoyed reading it. I also believe that this could be an interesting new diagnostic, potentially applicable to the interpretation also of other incubation studies looking at the fate of labelled substrates in soils. Nevertheless, I still have some open questions as outlined below.*

Thanks for these positive comments! we agree that the methodology can be adapted to other contexts where the goal is to study the pathway of transformation (stabilization as done here, or partitioning of carbon between microbial growth and respiration etc.) instead of the speed of the biogeochemical reactions.

*Since not being a mathematician, I am wondering how sensitive the model output is towards violating some of the assumptions made when solving the equation. One is the assumption that microbial biomass is in quasi-steady state. This might apply for individual states of the model, but not when integrating across the decomposition continuum? Since there is no input to the model, microbial biomass is first building up after residue addition (growth > mortality) and then declining (growth < mortality) during the subsequent starvation phase. Both phases are required to reconcile the studied decomposition patterns.*

Reviewer #2 had a similar comment regarding our assumption that microbes at quasi-equilibrium, so we respond to both comments in the same way.

The reviewer is correct that in these experiments microbial biomass likely increases first and then decreases when the substrate is consumed. We confirmed this trend by modelling (numerically) both substrates and microbial biomass in residues + POM and in MAOM (Eq. 1-2 and 5-6 in the main text), as shown in Figures R1 and R2 (denoted as 'full model'). For these model runs, we selected parameters reflecting the findings in the submitted manuscript and thus representative of the soil incubations we analyzed, but—different from the analytical model presented in the manuscript—we also had to specify kinetic parameters for residues + POM and microbial biomass. For the former we assumed residues + POM with a 6 month turnover time (reasonable for labile residues), whereas for the latter we considered two turnover rates (6 or 2 months, consistent with the range estimated for surface soils by Spohn et al., 2016). With the lower microbial mortality, there are some discrepancies between the full model and the minimal model presented in the manuscript—specifically, C accumulates in microbial biomass before being released and stabilized in MAOM in the full model, different from the fast transfer of C to MAOM in the minimal model (Figure R1B). This correspond to lower curves in the phase space—i.e., lower C in MAOM for a given C in residues + POM (Figure R1C). Thus, in the presence of microbial biomass with relatively slow turnover, we might be overestimating MAOM accumulation. However, this error decreases with faster microbial turnover (Figure R2A, B), or with slower decomposition rate of residues + POM (not shown). We can thus conclude that our approximation that microbes are in quasi-equilibrium is reasonable except when both microbial turnover is slow and residue turnover is fast.

While conducting this analysis, we also compared the analytical solution of the model in phase space to the numerical solution obtained by plotting the numerically simulated time trajectory of C in MAOM as a function of C in residues + POM. As expected, the two results are undistinguishable (black and gray dashed curves in Figure R1C and R2C). Adding these figures in a revised manuscript does not seem necessary, but we would include this discussion on the quasi-equilibrium approximation in a new section of the Discussion "4.2. Model limitations." In this new section, we would explain: "The model was constructed with five compartments (including POM and MAOM substrates and microbial biomass, as well as DOM), but assuming that microbial biomass and DOM are at quasi-equilibrium allows reducing the model to two compartments. This simplification has minor consequences on the POM and MAOM dynamics as long as both microbial biomass and DOM turn over faster than the POM and MAOM substrates. Microbial biomass has a turnover time in the order of a few months (Spohn et al., 2016) and DOM dynamics are even faster—shorter than the turnover of POM and MAOM. Therefore, our quasi-equilibrium assumption appears to be reasonable."

[Figure]

**Figure R1. Temporal trends in the fractions of added C in: A) residues + POM (microbial biomass, $c_{PB}$, and total, $c_P$) and B) MAOM (microbial biomass, $c_{MB}$, and total, $c_M$). C) Phase space representation of the same time trajectories—i.e., $c_M$ is plotted as a function of $c_P$. In all panels, curves refer to the 'full model' where microbial biomass dynamics are included (solid lines), and the 'minimal model' where microbial biomass is assumed in quasi-equilibrium (dashed lines). In panel C, the minimal model is solved both numerically (black dashed line) and analytically as in the submitted manuscript (gray thick dashed line). Parameters: $b$=1, $m$=1, $l$=0.1, $e$=0.2, $\kappa$=0.05; we also assumed first order kinetics for residues + POM decomposition, with decay constant of 2 $y^{-1}$, and for microbial mortality, with rate constant of 2 $y^{-1}$.**

[Figure]

**Figure R2. Same as Figure R1, but with a microbial mortality rate constant of 6 y⁻¹.**

*Also, equation 8 suggests that the decomposition of the POM and the MAOM pool are proportional along the decomposition process (right?). Is that the case in the real world, or would it not change e.g. with increasing depletion of easily degradable compounds with increasing decomposition in the POM fraction, which is not necessarily paralleled in the MAOM fraction when this is indeed built up from microbial residues?*

Equation (8) (i.e., $D_M \approx \kappa D_P \frac{c_M}{c_P}$,) states that there is a proportionality between MAOM decomposition rate constant and residues + POM decomposition rate constant. This does not mean that the rates themselves are proportional. In fact, we assume that both decomposition rates (for MAOM, $D_M$, and for residues + POM, $D_P$) are linear functions of the carbon contents in those two compartments (for MAOM, $c_M$, and for residues + POM, $c_P$). By definition, the decomposition rate constants for these two rates are given by the rates divided by the carbon contents in the respective compartment (for MAOM, $\frac{D_M}{c_M}$, and for residues + POM, $\frac{D_P}{c_P}$).

The only assumption we make is that these rate constants are proportional. Thus, it is still possible for the decomposition rate of residues + POM to be at first larger than that of MAOM, and then lower when most of the residue carbon is already in the MAOM compartment. We can clarify this in the Methods (after Eq. (8)): "This assumption only implies a proportionality between the decay constants, while the actual rates will still be different depending on the relative abundance of C in residues + POM and MAOM."

In the second part of this comment, the reviewer correctly points out that the chemical characteristics of the carbon compartments differ through time. Indeed, we have not accounted for such a change. It is possible that the decay constant for the residues + POM decomposition decreases through time due to consumption of labile compounds, but to capture this effect we would need a model with more compartments and thus more parameters. In most datasets we used, POM and MAOM were regarded as lumped

compartments without chemical differentiation. Therefore, it would be difficult to parameterize a more complex model with these datasets. We can comment on this model limitation in a new Discussion section "4.2. Model limitations."

In this new section we would explain: "POM and MAOM contain compounds with contrasting chemical characteristics (depending on residue chemistry in POM and on the stabilization pathway in MAOM), but we neglected these chemical differences both to keep the model simple and because of limited data to parameterize more than one compartment for POM and one for MAOM. As a consequence, we also neglected the decreasing rates of decomposition through time as a result of accumulating recalcitrant compounds. However, we can expect that less decomposable compounds remain in both POM and MAOM, so that the ratio of the decay constants for these compartments (i.e., parameter $\kappa$) should remain relatively stable, which is the only assumption we need to make in our derivation. Therefore, neglecting chemical heterogeneity would have major consequences when predicting decomposition rates, but is likely less problematic when modelling POM and MAOM in phase space."

*Another assumption of the modelling approach is that ex-vivo and in-vivo formed MAOM have the same decomposition. Since they will differ in chemical composition and CN ratio, this might not necessarily be the case as also discussed by the authors in lines 545ff and could also have biased the results.*

This is also a good point, but as explained above, with the current data we could not test this very reasonable hypothesis. We would explain our rationale for chemically lumped compartments in the new section "4.2. Model limitations"—please see our response above.

*If I understand it correctly, parameter fitting was not done based on total carbon and nitrogen in the two pools (POM and MAOM) of the studied soils, but just the fate of labelled residues added and their transfer to the MAOM fraction was considered, right? Correlations to bulk soil carbon concentrations of the original samples were done afterwards to see how the fitted parameters were affected by background SOC?*

Yes, the reviewer is correct. Total SOC is used to characterize the soil environment, based on the idea that different SOC contents might affect the contributions of different stabilization pathways via e.g., saturation of mineral surfaces or maintenance of a more or less active microbial community. We can clarify this point in the Methods section 2.2.2: "The model was fitted to residue-derived C and N contents in both residues + POM and MAOM fractions." Moreover, we can specify in section 2.2.3 that SOC is regarded as an index of overall C availability.

*Nitrogen will probably play an important role for model parameterization, as the in vivo pathway leads to higher MAOM-N contents than the ex vivo path. Some of the datasets seem to have contained information on both, nitrogen and carbon, others not (table S1). How were the different data-streams then used for model fitting?*

Too few datasets contained time series of N contents in POM and MAOM, so we could not use individual N datasets for parameter estimation, as we did for the C data. However, we used N data in Figure 3 to provide a general constraint on the model parameters. We also provide some examples of model fitting in Figure 4, but without using the estimated parameters in further analyses. This can be clarified in the Methods section 2.2.2: "Too few datasets included residue-derived N contents in soil fractions for a systematic analysis, so model parameters were fitted only to the C data, except for a few examples."

*I was surprised that Figure 3 shows a proportional decline of C and N in POM with decomposition, since my understanding was so far, based on litter decomposition experiments (which should be equivalent to POM decomposition), that C is lost in excess to N, leading to a relative enrichment of N versus C particularly during the early stages of litter decomposition (e.g. https://doi.org/10.1016/j.ejsobi.2018.02.003, https://doi.org/10.1007/s10021-004-0026-x). The concept of the present study and outcome of the model-data integration is now, that if litter or POM is decomposed next to minerals instead of in litter bags, nitrogen is transferred to the mineral phase as microbial necromass and not accumulating in the POM fraction. It would be nice, if the authors could elaborate on these different observations a bit more.*

We agree that this pattern is surprising. We are familiar with the CIDET dataset mentioned by the reviewer (Moore et al., 2006), and other empirical studies showed the same pattern (Parton et al., 2007). In our own work—though from a more theoretical perspective—we also found faster loss of C compared to N or P during litter decomposition (Manzoni et al., 2008, 2010). Yet, the data is clear, indicating that when mixed in the soil, residues lose C and N at rates proportional to their initial C:N ratio—i.e., with no preferential retention of N. As the reviewer suggests, from a purely stoichiometric perspective, this pattern could be explained by N-rich necromass being preferentially associated with soil minerals and thus being recovered in the MAOM fraction. This would be different from litterbag studies where necromass is recycled locally contributing to litter N enrichment during decomposition. We could elaborate these points at the end of Section 4.3 "Reconciling contrasting decomposition patterns in phase space."

*I was further wondering, how the transformation from organic to inorganic N and the potential stabilization of ammonium on minerals was considered in the cited experiments? If N accumulation on minerals by sorption as inorganic N is not considered,*

*could that bias the results in soils rich in clay minerals? Could this have contributed to the observation, that the in vivo pathway is more important in clay rich soils?*

The experiments did not distinguish between N stabilized in MAOM in organic vs. inorganic forms. Indeed, both processes might have occurred—stabilization of organic N or of mineral N from mineralization of the residues, even though inorganic N form are typically negligible when compared to the organic forms. The N from the added residues recovered in the MAOM fraction is the result of both processes. Therefore—if we understand the reviewer's comment correctly—there would be no bias due to some N remaining unaccounted for.

*It also surprised me, that the in vivo pathway should be more sensitive to saturation than the ex-vivo pathway (e.g. line 561). Microbes can also live and die directly attached to mineral surfaces. Probably more microbes can potentially live on mineral surfaces if these provide more substrate at higher OM-loading, so I am wondering what the mechanism behind the expected higher sensitivity of the in vivo pathway to saturation should be.*

Thanks for this curiosity. While there are no definitive explanations for this finding, it is consistent with the current understanding that the organic matter directly bonded to minerals is N-(and P-)rich (Spohn, 2024), and therefore more likely of direct microbial origin. This would be the organic matter susceptible to saturation, being dependent on the active mineral surface availability. The C-rich organic matter (i.e., more likely derived from the ex-vivo pathway) appears to be indirectly bonded to minerals, via organic matter-organic matter interactions (Spohn, 2024). This "vertical structure" of OM on minerals is advocated to not be constrained by saturation (Begill et al., 2023). We can summarize these arguments in Section 4.5: "This finding is consistent with N-rich organic matter—likely of microbial origin—directly bonding to minerals (Spohn, 2024) and thus being dependent on the availability of active mineral surfaces. In contrast, C-rich organic matter from the ex-vivo pathway tends to indirectly bond to minerals through organic matter-organic matter interactions (Spohn, 2024) and is thus less constrained by saturation of the mineral surfaces (Begill et al., 2023)."

*The results presented in Figures 5 and 6 regarding the relation between microbial CUE and CN ratios seem contradictory: while it is positive in Figure 5, it is negative in Figure 6. What is the difference? This should be clarified, since in the discussion the focus is on the observed negative relation (line 565 ff).*

The change in the slope from positive in the univariate regression to negative in the multivariate regression may be caused by i) the effects of data grouping (the multivariate regression is performed using a linear mixed effect model, with data source as a random factor) and ii) the effects introduced by interactions involving residue C:N (even though such effects are not statistically significant). This comment made us realize that the univariate relations might be confusing, because they offer an incomplete view on the actual drivers of each parameter. Therefore, we would remove them from a revised manuscript, leaving only the box plots (left column in Figure 5) to illustrate the parameter ranges and distributions. The statistical analysis would then focus on the more complete linear mixed effect model. Moreover, data points could be added in Figure 6B-C, to show the spread of the data in a scatter-plot format.

*It was not clear to me, why the authors always use the term "residues + POM" throughout the manuscript – what is the difference? Does the term "residue" stand for the labelled material added and traced into MAOM? Why is the same distinction then not also necessary for MAOM when it is composed of both, residue- and POM-derived OM?*

We used the term "residues + POM" to indicate C or N originated from the added residues that is either still in the form of "residue" (fragments larger than 2 mm or separated by hand) or POM (obtained via size or density fractionation). In many datasets, residues were not removed prior to fractionation, so it was not possible to distinguish between residues and POM. Therefore, we chose to model residues and POM combined, and MAOM in a separate compartment. We can clarify in Section 2.1.1 "The choice of merging these two fractions in one model compartment is motivated by the fact that in many datasets residues and POM were not separated." Further details on the type of data used and data processing are given in Section 2.2.1.

*In line 19 of the abstract, the authors state that MAOM is fuelled by microbes decomposing POM – since there were two microbial pools in the model – can a distinction be made how much MAOM is from each microbial pool? Similar to direct input by microbes in the MAOM, also plant/ex vivo-derived MAOM is subsequently recycled in microbial biomass – is this not also a kind of in vivo OM transformation, since it will also affect MAOM CN ratios?*

The statement in the Abstract is accurate, as the flow of C from residues + POM to MAOM is dominated by the in vivo pathway according to our results. Within MAOM there is indeed recycling of microbial necromass, but that does not contribute to MAOM accumulation—only to C and N cycling within the MAOM compartment. With time, more and more of the MAOM will be composed of necromass from MAOM microbes. In a sense we agree that this is also an in vivo transformation, but we would use this term to refer specifically to stabilization into MAOM of necromass from microbes decomposing residues + POM.

Mathematically, the fraction of MAOM originated from either one or the other microbial group could be calculated, but we are not sure it would add to our discussion about the transfer of C and N from residues + POM to MAOM.

*Line 539: remove "tend" or adapt accordingly*

Thanks, that was a leftover from a text edit.

**References**

Begill, N., Don, A., Poeplau, C., 2023. No detectable upper limit of mineral-associated organic carbon in temperate agricultural soils. Global Change Biology. doi:10.1111/gcb.16804

Manzoni, S., Jackson, R.B., Trofymow, J.A., Porporato, A., 2008. The global stoichiometry of litter nitrogen mineralization. Science 321, 684–686.

Manzoni, S., Trofymow, J.A., Jackson, R.B., Porporato, A., 2010. Stoichiometric controls dynamics on carbon, nitrogen, and phosphorus in decomposing litter. Ecological Monographs 80, 89–106.

Moore, T.R., Trofymow, J.A., Prescott, C.E., Fyles, J., Titus, B.D., 2006. Patterns of carbon, nitrogen and phosphorus dynamics in decomposing foliar litter in Canadian forests. Ecosystems 9, 46–62.

Parton, W., Silver, W.L., Burke, I.C., Grassens, L., Harmon, M.E., Currie, W.S., King, J.Y., Adair, E.C., Brandt, L.A., Hart, S.C., Fasth, B., 2007. Global-scale similarities in nitrogen release patterns during long-term decomposition. Science 315, 361–364.

Spohn, M., 2024. Preferential adsorption of nitrogen- and phosphorus-containing organic compounds to minerals in soils: A review. Soil Biology and Biochemistry 194, 109428. doi:10.1016/j.soilbio.2024.109428

Spohn, M., Pötsch, E.M., Eichorst, S.A., Woebken, D., Wanek, W., Richter, A., 2016. Soil microbial carbon use efficiency and biomass turnover in a long-term fertilization experiment in a temperate grassland. Soil Biology and Biochemistry 97, 168–175. doi:10.1016/j.soilbio.2016.03.008

---

## Author Response (AR1)

We thank associate editor Sara Vicca and the two reviewers for their comments. Our responses are reported below in normal font, while the editor's and reviewers' comments are in *italic*.

**Responses to the associate editor**

*Thank you for your clear responses to the reviewers. The three reviewers were generally positive about your work and mainly highlighted aspects that need further clarification. They also raised several questions out of curiosity. Your responses indicate that you can address most of these comments and questions. I am inviting you to submit a revised manuscript in which the reviewer comments have been addressed, along with a response letter with point-by-point answers to the reviewer comments. The revised manuscript will be returned to at least one of the previous reviewers.*

Our response to the reviewers' comments are reported below, focusing on the actual changes made in the revised manuscript. We have left extended discussions in the individual responses published during the open review phase. In addition to addressing the reviewers' comments, we also edited the text elsewhere to improve clarity and consistency in the use of symbols and terms.

**Responses to Reviewer #1**

*This study proposes a phase-space model fitted to about 40 studies which estimates that ~75% of MAOM C and nearly all of MAOM N are stabilized by the in vivo pathway, in contrast to measurements which tend to be lower than this. The authors posit a few reasons for the discrepancy, including ex vivo stabilization pathways missing from the model, and the most interesting which is that necromass may be stabilized faster but turns over at a faster rate, resulting in less persistent MAOM from necromass.*

We thank the reviewer for their feedback and we agree that the finding of dominant in vivo stabilization is interesting and complementary to recent literature.

*I think in order to explore the hypotheses brought up by this study it is useful to run the model considering time. I assume that you would get the same curves that can be seen in the main results (Figs 2 and 4) when running the model forward as differential equations and plotting the model output as the relationship between pools, so I'm not sure what the phase space simplification really adds.*

Solving the mass balance equations through time and then plotting one state variable as a function of the other will give the same results as directly solving the system in the phase space (see an example in the published response to Reviewer #2). We now clarify why we move from time domain to phase space at the beginning of Section 2.1.3:

"Equations (4) and (7) can be solved through time after specifying how the rates $D_P$ and $D_M$ vary with the state variables $C_P$ and $C_M$, as well as with environmental conditions. To remove part of the variability induced by environmental conditions and to limit the number of model parameters, we move from this representation in the time domain to one in the phase space."

We have revised Section 3.3 to refer to the new Figure 4, which now includes examples of time trajectories. We also discuss the advantages of a phase space representation in Section 4.1, were we added:

"This is particularly useful when the temporal trajectories are very different among datasets (e.g., Figure 4A-F), while in the phase space data start exhibiting more consistent (and simpler) trends (Figure 4G-I)."

*I think that retaining the ability to consider absolute values of MAOM and POM and track losses (C and N min) are relevant for addition scenarios in the field where total C retained in the system is as or more relevant than the form of C.*

Normalizing the C and N contents in the modelled pools by the added residue C and N does not preclude the possibility to calculate absolute amounts of C and N in the soil pools, as we explained just before Eq. (10) where the normalization step is presented:

"Moreover, the equations expressed in normalized form are independent of the units used to quantify inputs and mass in each compartment, and if needed it is easy to convert the normalized variables into absolute quantities by multiplying by the mass of added residue C."

*There were similarly some places in the text (L357-359) and Fig. 4 where I was really curious how accumulation of N vs C in MAOM was evolving over time.*

In the revised manuscript, we expanded Figure 4 to also illustrate the time series of the data, as suggested by the reviewer. Moreover, we added Appendix A (with a new figure) to show how C and N accumulate in MAOM at first and then are lost. In the Appendix A we explain:

"General trends in MAOM accumulation can be assessed by calculating the change in C or N in MAOM per unit change in residues + POM ($\Delta c_M \Delta c_P^{-1}$). Because the residues + POM compartment loses mass due to decomposition, its changes are always negative ($\Delta c_P < 0$). As a consequence, negative relative changes in MAOM ($\Delta c_M \Delta c_P^{-1} < 0$) indicate accumulation of mass in MAOM. Generally, in the early phase of decomposition when the fraction of remaining residues + POM is still high, both C and N accumulated in MAOM, but below a $c_P$ or $n_P$ threshold, both C and N are lost from MAOM (Fig. A1). The turning points when MAOM starts being depleted are at $c_P \approx 0.26$ and $n_P \approx 0.18$."

Appendix A is referred to in the main text, at the beginning of Section 3.1:

"In general, the data show that C and N from the residues + POM compartment accumulate in MAOM in the early decomposition phase, while later both residues + POM and MAOM compartments lose mass (Appendix A)."

*I suppose my ask to the authors is to justify to readers the utility of phase space modeling compared to running the time-dependent form of the model, or to use time-dependent modeling to more completely address some of the hypotheses advanced, especially about N vs C accumulation in MAOM.*

We agree with the reviewer that the temporal dynamics contain additional information, but that will mostly reflect incubation conditions rather than the intrinsic capacity of the soil and its microbial community to promote organic matter stabilization in MAOM through the in vivo or ex vivo pathway. We have revised the manuscript as suggested above to better motivate our choice to focus on the phase space rather than the time domain, but we would be reluctant to fit the model to the time series as interpreting patterns in both kinetic constants (which we now do not consider) and partitioning coefficients (our current focus) would be difficult and results would depend on the selected kinetics.

*L59-60: I think this sentence needs a main clause verb.*

Thanks for catching this mistake. Deleting "and" fixes the problem.

*L120: Probably good to acknowledge that some C will be lost as DOC leaching, even if it is small and you don't consider it here.*

We clarified by adding that "Leaching of dissolved organic matter is neglected."

*Fig 1: A fraction of necromass is sorbed and the rest goes back into particulate C... what is that fraction based on?*

The fraction of necromass that is sorbed is quantified by the parameter *m*, which is estimated by fitting the model to the data. Therefore, we do not make any assumptions about this fraction, but rather let the data tell if it is large or small. We would not add a clarification on this comment, because the estimation approach for parameter *m* is presented in Section 2.2.2.

*At L322, Section 0 is referenced as where you explain how m is constrained but I could not find this section.*

Sorry about this, there was an issue with a section referencing. It should read "Section 3.2"

*Eq.10 is the same as Eq. 9 but with a different boundary condition. I would suggest referencing Eq. 9 instead of printing the equation again.*

Equation (10) looks similar, but besides the different boundary conditions, it also includes normalized variables (small *c* instead of capital *C*), so we would prefer to keep it.

*N mineralization assumes a one-way flow where N is mineralized but not taken up again. This is similar to a lot of soil models but not necessarily realistic, so perhaps worth acknowledging that in text somewhere.*

We understand the concern, but the reviewer missed that this is not necessarily the case in our formulation of net N mineralization, which accounts for possible net N immobilization when microbial N demand is higher than the N supply from organic matter decomposition. We can add a clarification below Eq. (17):

"These formulations for net N mineralization allow capturing both net N release if substrates are sufficiently rich in N ($N_{PS}/C_{PS} > e_P r_{PB}$, $N_{MS}/C_{MS} > e_M r_{MB}$) and net N immobilization when they cannot provide enough N for microorganisms ($N_{PS}/C_{PS} < e_P r_{PB}$, $N_{MS}/C_{MS} < e_M r_{MB}$)."

*Table S1: If POM+residues are considered as one category in the model (Fig 1) then why are they separated out here? Please clarify how each category is allocated when used in the model.*

The dataset contains data on residues alone, POM alone, or combined residues + POM. As most datasets report combined residues + POM, we chose to use this as a state variable in the model. We added a short explanation in Section 2.1.1:

"The choice of merging residues and POM in one model compartment is motivated by the fact that in many datasets they were not separated."

*L266 seems to imply that those studies without POM+residues reported were not used, does this change the reported sample size used in analyses?*

Six studies reporting only POM or only residue C were not used, so only 36 of the 42 studies included in the database were actually used in this work. We clarified in Section 2.2.1:

"Due to these gaps, 36 out of the total 42 datasets are used in the following analyses"

While all data from these 36 studies was used in the first part of the Results, only a subset of the data was used for model fitting, as explained in Section 2.2.2.

*L357-359: I found it curious that a higher C:N ratio causes lower C accumulation in MAOM but higher N accumulation. Does the C:N of MAOM really decrease in this scenario or is this an artifact of only viewing Cm as relative to Cp which is also changing over time?*

Higher residue C:N does not affect C accumulation in MAOM (dashed and solid curves overlap in Figure 4, panel in the top row, second column); however, higher C:N does promote N accumulation in MAOM because more N is immobilized by microorganisms, converted into necromass, and eventually transferred from the residues + POM to the MAOM compartment. We added a minor clarification in Section 3.3:

"… relatively N poor residues exhibit strong N immobilization and accumulation of N in MAOM"

*In Figure 4, it is also interesting that N seems to accumulate faster than C in MAOM, but again we are not looking at time here so travel along either curve could occur at different speeds. Can you discuss this or provide some information about the time component?*

The reviewer is correct—we are not looking at temporal trajectories, so it is not possible to read reaction speed in these phase spaces. As suggested above, we added panels in Figure 4 to illustrate the temporal changes and commented on those patterns in Section 3.3.

*Following from the understanding that in vivo is more important for N than C, I wonder if this can tell us something about the rate of the in vivo pathway vs ex vivo.*

We appreciate the reviewer curiosity, but as explained above with the current analysis, we cannot evaluate and rank the rates of C and N stabilization, but only the relative proportions.

*Figure 3b: I know there are many data sources, but can the data source color bar be discretized?*

We replace the existing Figure 3B with a revised version where the colorbar for the data source is discretized.

*L466: I was a little surprised to see such a strong prediction of clay trends in B and C but no trend in univariate space (Fig 5). Do you feel confident in the LME prediction?*

P-values for the clay effects on the fraction of C and N following the in vivo pathway are low (<0.05) and coefficients are relatively large, so yes, we are confident there is a trend. If were hoping to strengthen our analysis by adding some datasets we found during the review phase (Hilscher and Knicker, 2011; Córdova et al., 2018), but such datasets were deemed not suitable (either biochar incubations, while the other studies focused on fresh residues, or experimental design incompatible with the model assumptions). However, we did add data from an unpublished study (Cotrufo et al., in review).

To address this comment and also a similar one by Reviewer #3, we removed the univariate analysis.

**Responses to Reviewer #2**

*Manzoni and Cotrufo propose a two-compartment model (POM + residues and MAOM), which can be used to diagnose through which pathway (in vivo vs. ex vivo) C and N are stabilized. The simplicity of the model reduces the number of parameters and allows the authors to solve their equations analytically in the so-called space phase, i.e. they focus on relations between compartments instead of analyzing temporal trajectories to detect underlaying mechanisms. The model suggests that stabilization via the in vivo pathway is more relevant in clay-rich soils and less relevant in C-rich soils, whereby the calculated fractions seem to be higher than reported by earlier studies. Overall, the paper is well (and lively!) written and the chosen approach is an interesting method to explain potentially appearing inconsistencies in data sets. E.g. the model is able to capture MAOM stabilization during early decomposition and destabilization of MAOM in late decomposition by the same mechanism. From that perspective and because it adds a new insight into the process of C and N stabilization, the paper is a highly relevant for the BG community. Also, discrepancies with earlier studies are mentioned and discussed appropriately.*

We thank the reviewer for their supportive comments—both on our writing style and on the content of the manuscript.

*However, I'm concerned about the assumptions and approximations, which are made during the derivation of the solution, and that are a bit loosely justified (see below), and would ask the authors to elaborate these parts a bit further.*

We agree that assumptions need to be spelled out clearly—please see our responses below.

*As mentioned before, I'd ask the authors to explain used assumptions and approximations in more detail. Especially the assumption of (quasi-)equilibrium needs more explanation and discussion. Why (under which circumstances) can quasi-equilibrium be assumed?*

In our published response to Reviewers #2 and #3 we tested numerically the quasi-steady state assumption and confirmed that it is reasonable in most conditions except when both microbial turnover is slow and residue turnover is fast. We have now commented on the quasi-equilibrium approximation in the new section of the Discussion "4.2. Model limitations":

"The model was initially constructed with five compartments (including POM and MAOM substrates and microbial biomass, as well as DOM), but assuming that microbial biomass and DOM are at quasi-equilibrium allows reducing the model to two compartments. This simplification has minor consequences on the POM and MAOM dynamics as long as both microbial biomass and DOM turn over faster than the POM and MAOM substrates. Microbial biomass has a turnover time in the order of a few months (Spohn et al., 2016) and DOM dynamics are even faster—shorter than the turnover of POM and MAOM. Therefore, our quasi-equilibrium assumption appears to be reasonable."

*Are the data filtered by that?*

No, there is no way from the data available to filter datasets according to the turnover rate of microbial biomass or DOC. Our arguments for model simplification are general and not data-driven.

*How does this relate to the later discussion about "early" and "late" decomposition?*

Early and late decomposition phases are discussed based on residues + POM decomposition and MAOM accumulation and subsequent decomposition, which are both evident from the data and model results even when assuming that microorganisms are in quasi-equilibrium.

*How do this assumption and the other assumptions and approximations, which are used to derive the final analytically-solvable model, i.e. the first-order decay with a constant decomposition rate or the "similar" C:N ratios, affect model results and limit their transferability, e.g. under a changing climate?*

We would like to first clarify that we did not assume first-order decay, but only that the decay rate constants for residues + POM and MAOM decomposition are proportional. This assumption is less strict than assuming first-order decay for these processes. Regarding our assumption of a single value for microbial biomass C:N ratio, we could not find evidence in the literature of different microbial C:N between POM and MAOM fractions. We can elaborate on these model assumptions in the new Discussion section "4.2. Model limitations":

"Microorganisms growing on residues + POM are likely different from those feeding on organic matter desorbed from minerals. For example, we could expect higher fungal to bacterial ratio in POM, with higher microbial biomass C:N and possibly lower CUE (Soares and Rousk, 2019), but lack of specific information on microbial traits within the soil fractions does not allow us to parameterize these communities in the model (though soil fraction-specific traits are retained in the general solutions of the mass balance equations)."

*Figure 2: I think, it would help the reader, if you could add a reference figure (base values used) to understand the overall model behavior and the direction of the changes. I would also like to see the varied values, i.e. the values that are used to derive the solid and dashed lines in each sub-figure, in the figure description.*

We have revised Figure 2 according to this suggestion, by adding a column with model results obtained using the baseline parameter values. The values of the parameters used to make each panel are listed in the legends at the bottom of the figure, so it seems redundant to also report them in the caption.

*What is "Section 0"? (L322, L416)*

These are issues with section referencing. They should read "Section 3.2" and "Section 3.4", respectively.

*L34 Please check the unit.*

Font issue—it should be µm.

*Table 2: I personally would like to have the Parameter/Symbol entries all centered (or maybe all right-bounded), but not changing within lines, which seems to happen for all entries that do not have subscripts. Or is there I reason for that?*

No particular reason, but it seems equation objects sit in the center by default despite normal text being aligned to the left. The table has now been formatted so that all symbols are centered.

**Responses to Reviewer #3**

*The submitted manuscript aims at improving our understanding of the formation of mineral associated organic matter (MAOM), particularly the relative contributions of plant versus microbial derived carbon to MAOM formation. Therefore, they apply the innovative approach of fitting respective parameters of a simple soil microbial model in phase space. The datasets used are from decomposition experiments, where added labelled residues are traced from the particulate organic matter (POM) to the MAOM fraction. Instead of looking at temporal changes, the relative proportional changes of POM and MAOM along the decomposition continuum are mathematically formulated and solved, considering both C- and N- dynamics. The results showed that the in-vivo pathway (microbial derived) was more important than the ex-vivo pathway (plant-derived) MAOM-C and particularly MAOM-N formation. It was further possible to identify some controls on the relative importance of the two pathways. The authors conclude that the in vivo-pathway is particularly important in clay rich and carbon poor soils and that this path is accordingly stronger controlled by available mineral surfaces. The authors did a great job in writing and explaining each modelling step in the manuscript and I enjoyed reading it. I also believe that this could be an interesting new diagnostic, potentially applicable to the interpretation also of other incubation studies looking at the fate of labelled substrates in soils. Nevertheless, I still have some open questions as outlined below.*

Thanks for these positive comments! we agree that the methodology can be adapted to other contexts where the goal is to study the pathway of transformation (stabilization as done here, or partitioning of carbon between microbial growth and respiration etc.) instead of the speed of the biogeochemical reactions.

*Since not being a mathematician, I am wondering how sensitive the model output is towards violating some of the assumptions made when solving the equation. One is the assumption that microbial biomass is in quasi-steady state. This might apply for individual states of the model, but not when integrating across the decomposition continuum? Since there is no input to the model, microbial biomass is first building up after residue addition (growth > mortality) and then declining (growth < mortality) during the subsequent starvation phase. Both phases are required to reconcile the studied decomposition patterns.*

In our published response to Reviewers #2 and #3 we tested numerically the quasi-steady state assumption and confirmed that it is reasonable in most conditions except when both microbial turnover is slow and residue turnover is fast. We have now commented on the quasi-equilibrium approximation in the new section of the Discussion "4.2. Model limitations":

"The model was initially constructed with five compartments (including POM and MAOM substrates and microbial biomass, as well as DOM), but assuming that microbial biomass and DOM are at quasi-equilibrium allows reducing the model to two compartments. This simplification has minor consequences on the POM and MAOM dynamics as long as both microbial biomass and DOM turn over faster than the POM and MAOM substrates. Microbial biomass has a turnover time in the order of a few months (Spohn et al., 2016) and DOM dynamics are even faster—shorter than the turnover of POM and MAOM. Therefore, our quasi-equilibrium assumption appears to be reasonable."

*Also, equation 8 suggests that the decomposition of the POM and the MAOM pool are proportional along the decomposition process (right?). Is that the case in the real world, or would it not change e.g. with increasing depletion of easily degradable compounds with increasing decomposition in the POM fraction, which is not necessarily paralleled in the MAOM fraction when this is indeed built up from microbial residues?*

Equation (8) (i.e., $D_M \approx \kappa D_P \frac{C_M}{C_P}$,) states that there is a proportionality between MAOM decomposition rate constant and residues + POM decomposition rate constant. This does not mean that the rates themselves are proportional. By definition, the decomposition rate constants for these two rates are given by the rates divided by the carbon contents in the respective compartment (for MAOM, $\frac{D_M}{C_M}$, and for residues + POM, $\frac{D_P}{C_P}$). The only assumption we make is that these rate constants are proportional, so it is still possible for the decomposition rate of residues + POM to be at first larger than that of MAOM, and then lower when most of the residue carbon is already in the MAOM compartment. We can clarify this in Section 2.1.2 (just after Eq. (8)):

"This assumption only implies a proportionality between the decay constants, while the actual rates will still be different depending on the relative abundance of C in residues + POM and MAOM."

In the second part of this comment, the reviewer correctly points out that the chemical characteristics of the carbon compartments differ through time. Indeed, we have not accounted for such a change. We can comment on this model limitation in the new Discussion section "4.2. Model limitations":

"Residues + POM and MAOM contain compounds with contrasting chemical characteristics (depending on residue chemistry and on the pathway of stabilization into MAOM, respectively), but we neglected these chemical differences both to keep the model simple and because of limited data to parameterize more than one compartment for residues + POM and one for MAOM. As a consequence, we also neglected the decreasing rates of decomposition through time as a result of accumulating recalcitrant compounds. However, we can expect that less decomposable compounds remain in both POM and MAOM (because of their different chemical recalcitrance or accessibility, respectively), so that the ratio of the decay constants for these compartments (i.e., parameter ☐) should remain relatively stable, which is the only assumption we need to make in our derivation. Therefore, neglecting chemical heterogeneity may significantly affect the prediction of decomposition rates, but it is likely to be less important when modelling residues + POM and MAOM in phase space."

*Another assumption of the modelling approach is that ex-vivo and in-vivo formed MAOM have the same decomposition. Since they will differ in chemical composition and CN ratio, this might not necessarily be the case as also discussed by the authors in lines 545ff and could also have biased the results.*

This is also a good point, but as explained above, with the current data we could not test this very reasonable hypothesis. We would explain our rationale for chemically lumped compartments as explained above.

*If I understand it correctly, parameter fitting was not done based on total carbon and nitrogen in the two pools (POM and MAOM) of the studied soils, but just the fate of labelled residues added and their transfer to the MAOM fraction was considered, right? Correlations to bulk soil carbon concentrations of the original samples were done afterwards to see how the fitted parameters were affected by background SOC?*

Yes, the reviewer is correct. Total SOC is used to characterize the soil environment, based on the idea that different SOC contents might affect the contributions of different stabilization pathways via e.g., saturation of mineral surfaces or maintenance of a more or less active microbial community. We can clarify this point in the Section 2.2.2:

"The model was fitted to residue-derived C and N contents in both residues + POM and MAOM fractions."

Moreover, we also clarified in Section 2.2.3 that SOC is regarded as an index of overall C availability.

*Nitrogen will probably play an important role for model parameterization, as the in vivo pathway leads to higher MAOM-N contents than the ex vivo path. Some of the datasets seem to have contained information on both, nitrogen and carbon, others not (table S1). How were the different data-streams then used for model fitting?*

Too few datasets contained time series of N contents in POM and MAOM, so we could not use individual N datasets for parameter estimation, as we did for the C data. This limitation has now been spelled out in Section 2.2.2:

"Too few datasets included residue-derived N contents in soil fractions for a systematic analysis, so model parameters were fitted only to the C data, except for a few examples."

*I was surprised that Figure 3 shows a proportional decline of C and N in POM with decomposition, since my understanding was so far, based on litter decomposition experiments (which should be equivalent to POM decomposition), that C is lost in excess to N, leading to a relative enrichment of N versus C particularly during the early stages of litter decomposition (e.g. https://doi.org/10.1016/j.ejsobi.2018.02.003, https://doi.org/10.1007/s10021-004-0026-x). The concept of the present study and outcome of the model-data integration is now, that if litter or POM is decomposed next to minerals instead of in litter bags, nitrogen is transferred to the mineral phase as microbial necromass and not accumulating in the POM fraction. It would be nice, if the authors could elaborate on these different observations a bit more.*

We agree that this pattern is surprising and we elaborated on these points at the end of Section 4.3:

"The phase space representation also shows that the stoichiometry of residues + POM is conserved during residue decomposition and stabilization, regardless of the residue initial C:N ratio (Figure 3A). This result might seem surprising, as N is preferentially retained during residue decomposition, often resulting in a temporary net N accumulation if residues are N poor (Moore et al., 2006; Parton et al., 2007). This pattern can be explained by microbial N immobilization and recycling of N rich microbial necromass within the residues, which gradually lowers the residue C:N to values close to those of microbial biomass (Manzoni et al., 2008). The observation that the residues + POM compartment retains the initial residue C:N ratio indicates that microbial necromass is not recycled within that compartment, but rather it is stabilized into the MAOM fraction through the in vivo pathway (parameter m=1). Therefore, the phase space of C and N in residues + POM both informs about stabilization mechanisms and helps constrain model parameters."

*I was further wondering, how the transformation from organic to inorganic N and the potential stabilization of ammonium on minerals was considered in the cited experiments? If N accumulation on minerals by sorption as inorganic N is not considered, could that bias the results in soils rich in clay minerals? Could this have contributed to the observation, that the in vivo pathway is more important in clay rich soils?*

The experiments did not distinguish between N stabilized in MAOM in organic vs. inorganic forms. Indeed, both processes might have occurred—stabilization of organic N or of mineral N from mineralization of the residues, even though inorganic N form are typically negligible when compared to the organic forms. The N from the added residues recovered in the MAOM fraction is the result of both processes. Therefore—if we understand the reviewer's comment correctly—there would be no bias due to some N remaining unaccounted for.

*It also surprised me, that the in vivo pathway should be more sensitive to saturation than the ex-vivo pathway (e.g. line 561). Microbes can also live and die directly attached to mineral surfaces. Probably more microbes can potentially live on mineral surfaces if these provide more substrate at higher OM-loading, so I am wondering what the mechanism behind the expected higher sensitivity of the in vivo pathway to saturation should be.*

Thanks for raising this point. While there are no definitive explanations for this finding, we have summarized the arguments proposed in the recent literature in Section 4.5:

"This finding is consistent with N-rich organic matter—likely of microbial origin—directly bonding to minerals (Spohn, 2024) and thus being dependent on the availability of active mineral surfaces. In contrast, C-rich organic matter from the ex-vivo pathway tends to indirectly bond to minerals through organic matter-organic matter interactions (Spohn, 2024) and is thus less constrained by saturation of the mineral surfaces (Begill et al., 2023)."

*The results presented in Figures 5 and 6 regarding the relation between microbial CUE and CN ratios seem contradictory: while it is positive in Figure 5, it is negative in Figure 6. What is the difference? This should be clarified, since in the discussion the focus is on the observed negative relation (line 565 ff).*

To avoid confusion, and also in response to a comment by Reviewer #1, we have removed the univariate regressions, leaving only the box plots (left column in the original Figure 5) to illustrate the parameter ranges and distributions. The statistical analysis now focuses on the more complete linear mixed effect model. Moreover, data points have been added in Figure 6B-C, to show the spread of the data in a scatter-plot format.

*It was not clear to me, why the authors always use the term "residues + POM" throughout the manuscript – what is the difference? Does the term "residue" stand for the labelled material added and traced into MAOM? Why is the same distinction then not also necessary for MAOM when it is composed of both, residue- and POM-derived OM?*

We used the term "residues + POM" to indicate C or N originated from the added residues that is either still in the form of "residue" (fragments larger than 2 mm or separated by hand) or POM (obtained via size or density fractionation). In many datasets, residues were not removed prior to fractionation, so it was not possible to distinguish between residues and POM. Therefore, we chose to model residues and POM combined, and MAOM in a separate compartment. We clarified in Section 2.1.1:

"The choice of merging these two fractions in one model compartment is motivated by the fact that in many datasets residues and POM were not separated."

*In line 19 of the abstract, the authors state that MAOM is fuelled by microbes decomposing POM – since there were two microbial pools in the model – can a distinction be made how much MAOM is from each microbial pool? Similar to direct input by microbes in the MAOM, also plant/ex vivo-derived MAOM is subsequently recycled in microbial biomass – is this not also a kind of in vivo OM transformation, since it will also affect MAOM CN ratios?*

The statement in the Abstract is accurate, as the flow of C from residues + POM to MAOM is dominated by the in vivo pathway according to our results. Within MAOM there is indeed recycling of microbial necromass, but that does not contribute to MAOM accumulation—only to C and N cycling within the MAOM compartment. With time, more and more of the MAOM will be composed of necromass from MAOM microbes. In a sense we agree that this is also an in vivo transformation, but we would use this term to refer specifically to stabilization into MAOM of necromass from microbes decomposing residues + POM.

Mathematically, the fraction of MAOM originated from either one or the other microbial group could be calculated, but we are not sure it would add to our discussion about the transfer of C and N from residues + POM to MAOM.

*Line 539: remove "tend" or adapt accordingly*

Thanks, that was a leftover from a text edit.

**References**

Begill, N., Don, A., Poeplau, C., 2023. No detectable upper limit of mineral-associated organic carbon in temperate agricultural soils. Global Change Biology. doi:10.1111/gcb.16804

Córdova, S., Olk, D., Dietzel, R., Mueller, K., Archontouilis, S., Castellano, M., 2018. Plant litter quality affects the accumulation rate, composition, and stability of mineral-associated soil organic matter. Soil Biology & Biochemistry 125, 115–124. doi:10.1016/j.soilbio.2018.07.010

Cotrufo, F., Haddix, M.L., Mullen, J.L., Zhang, Y., McKay, J.K., in review. Deepening root inputs: A mod-ex study on soil carbon accrual potentials for maize.

Hilscher, A., Knicker, H., 2011. Degradation of grass-derived pyrogenic organic material, transport of the residues within a soil column and distribution in soil organic matter fractions during a 28 month microcosm experiment. Organic Geochemistry 42, 42–54. doi:10.1016/j.orggeochem.2010.10.005

Manzoni, S., Jackson, R.B., Trofymow, J.A., Porporato, A., 2008. The global stoichiometry of litter nitrogen mineralization. Science 321, 684–686.

Moore, T.R., Trofymow, J.A., Prescott, C.E., Fyles, J., Titus, B.D., 2006. Patterns of carbon, nitrogen and phosphorus dynamics in decomposing foliar litter in Canadian forests. Ecosystems 9, 46–62.

Parton, W., Silver, W.L., Burke, I.C., Grassens, L., Harmon, M.E., Currie, W.S., King, J.Y., Adair, E.C., Brandt, L.A., Hart, S.C., Fasth, B., 2007. Global-scale similarities in nitrogen release patterns during long-term decomposition. Science 315, 361–364.

Soares, M., Rousk, J., 2019. Microbial growth and carbon use efficiency in soil: Links to fungal-bacterial dominance, SOC-quality and stoichiometry. Soil Biology and Biochemistry 131, 195–205. doi:10.1016/j.soilbio.2019.01.010

Spohn, M., 2024. Preferential adsorption of nitrogen- and phosphorus-containing organic compounds to minerals in soils: A review. Soil Biology and Biochemistry 194, 109428. doi:10.1016/j.soilbio.2024.109428

---

## Referee Report (RR1)

Revision of "Mechanisms of soil organic carbon and nitrogen stabilization in mineral associated organic matter – Insights from modelling in phase space" by Manzoni and Cotrufo.

**General Comments**

Manzoni and Cotrufo responded clearly to the reviewers' questions and comments, and took them into account when revising the manuscript. They added respective parts to their manuscript and changed figures accordingly. This justifies assumptions and following derivations, which makes it easier to follow the methods, and clarifies the results. Mentioning and discussing critical points increases the reliability of the study. In cases, where the authors decided not to change the manuscript, the decisions are clearly explained. Overall, I'm satisfied with the work that the authors have done, especially with the text edits, but I wondered about some figure changes, which I report in more detail below.

**Text edits:**

I especially like the added section 4.2 about model limitations, which show that the authors considered the reviewers' concerns about the study and potential limitations, and I would like to highlight the manifold clarifications in section 2, which justify assumptions and thus support understanding the equations. Also, the replacement of "and" by "+" when speaking about POM + residues, which was not requested by any reviewer as far as I see, improves readability. Additionally, being more precise about the actual number of studies, which are used for each step, instead of mentioning a vague number (around 40), makes the study more reliable.

**Figure edits:**

The authors have made a huge effort in editing the figures in response to the reviewers' comments and suggestions, which in general improves the figures, simplifies their interpretation and strengthen their messages. I'm especially happy with the added baseline plots in figure 2 that help to understand general model behavior and thus to interpret changes, and the added time trajectory plots in figure 4, which clearly show the advantage of analyzing within the phase space instead of time trajectories. I also like the simplification of figure 5 by only showing the boxplots instead of all values, because this makes the message much clearer. However, I find the discretizing of data sources (fig. 3b, color bar) hard to see. Not sure, if it would help to either use a wider range of colors (e.g. by adding reddish colors, which may be problematic for color blind people), or by using discrete intervals (1-5, 6-10, 11-15,…) instead of having one color for each number, or to leave out numbers that are not used, e.g. if for example the values 2, 6, 14,… are not used, they could appear black, to save colors for numbers that are actually used. And I'm wondering about the changes in figure 6. E.g. the changed numbers in panel A, and the differences in SOC median (previous version: 0.016) and 50[th] percentile (revised version: 0.013) in panel B. However, my concerns may root in the fact that I have seen the previous version of the manuscript, but not prevent a new reader from getting the points, and I really like the visualization of the actual data points in panels B and C.

---

## Author Response (AR2)

**Responses to the associate editor**

*Your revised manuscript has been re-evaluated by one of the original reviewers, who was generally very positive about the revisions but did make a few minor comments regarding figures 3 and 6. Please consider these before uploading your final files (but do keep in mind the guidelines for colorblindness for Figure 3 - see https://www.biogeosciences.net/submission.html for more information).*

Our responses are reported below in normal font, while the editor's and reviewers' comments are in *italic*. The comments by Reviewer #2 have been taken into account, and we checked that the chosen color palettes followed the guidelines for colorblindness. We also made minor text edits in the Supplementary Information.

*To add to the reviewer suggestions for these figures:*

*- In figure 3, you may also consider reducing the size of the symbols to improve the visualization*

*- In figure 6 (B and C), it would be good to also write the variable for the color bar (SOC in B, C:N and C; similar to how you did this in Fig. 3).*

Symbol size was reduced in Figure 3 and colorbar labels were added in Figure 6. Thanks for these suggestions.

**Responses to Reviewer #2**

*General Comments: Manzoni and Cotrufo responded clearly to the reviewers' questions and comments, and took them into account when revising the manuscript. They added respective parts to their manuscript and changed figures accordingly. This justifies assumptions and following derivations, which makes it easier to follow the methods, and clarifies the results. Mentioning and discussing critical points increases the reliability of the study. In cases, where the authors decided not to change the manuscript, the decisions are clearly explained. Overall, I'm satisfied with the work that the authors have done, especially with the text edits, but I wondered about some figure changes, which I report in more detail below.*

*Text edits: I especially like the added section 4.2 about model limitations, which show that the authors considered the reviewers' concerns about the study and potential limitations, and I would like to highlight the manifold clarifications in section 2, which justify assumptions and thus support understanding the equations. Also, the replacement of "and" by "+" when speaking about POM + residues, which was not requested by any reviewer as far as I see, improves readability. Additionally, being more precise about the actual number of studies, which are used for each step, instead of mentioning a vague number (around 40), makes the study more reliable.*

Thanks for these supportive comments. We agree that the revised text is easier to follow and more accurate.

*Figure edits: The authors have made a huge effort in editing the figures in response to the reviewers' comments and suggestions, which in general improves the figures, simplifies their interpretation and strengthen their messages. I'm especially happy with the added baseline plots in figure 2 that help to understand general model behavior and thus to interpret changes, and the added time trajectory plots in figure 4, which clearly show the advantage of analyzing within the phase space instead of time trajectories. I also like the simplification of figure 5 by only showing the boxplots instead of all values, because this makes the message much clearer. However, I find the discretizing of data sources (fig. 3b, color bar) hard to see. Not sure, if it would help to either use a wider range of colors (e.g. by adding reddish colors, which may be problematic for color blind people), or by using discrete intervals (1-5, 6-10, 11-15,…) instead of having one color for each number, or to leave out numbers that are not used, e.g. if for example the values 2, 6, 14,… are not used, they could appear black, to save colors for numbers that are actually used.*

We have now added black lines to separate the colors in the colorbar, and reduced symbol size to improve visibility of both data points and colors. It should be noted that the source number has no particular meaning, as it refers to the 'source code' in the database, so grouping data sources would not be useful. Also, pretty much all data are shown in Figure 3B, so skipping some data source number would not help in the visualization. The interested reader can check individual datasets in the open-access database.

*And I'm wondering about the changes in figure 6. E.g. the changed numbers in panel A, and the differences in SOC median (previous version: 0.016) and 50th percentile (revised version: 0.013) in panel B. However, my concerns may root in the fact that I have seen the previous version of the manuscript, but not prevent a new reader from getting the points, and I really like the visualization of the actual data points in panels B and C.*

Median values changed slightly because we have added a dataset to our database, which resulted in additional data points in Figure 6.